# ZEB1/NuRD complex suppresses TBC1D2b to stimulate E-cadherin internalization and promote metastasis in lung cancer

Roxsan Manshouri[1,2], Etienne Coyaud[3], Samrat T. Kundu[1,2], David H. Peng[1,2], Sabrina A. Stratton [4], Kendra Alton [4], Rakhee Bajaj [1,2], Jared J. Fradette[1,2], Rosalba Minelli[5], Michael D. Peoples[5], Alessandro Carugo[6], Fengju Chen[7], Christopher Bristow[6], Jeffrey J. Kovacs[6], Michelle C. Barton [4], Tim Heffernan[6], Chad J. Creighton [7], Brian Raught[3] & Don L. Gibbons[1,2]*

Non-small cell lung cancer (NSCLC) is the leading cause of cancer-related death worldwide, due in part to the propensity of lung cancer to metastasize. Aberrant epithelial-to-mesenchymal transition (EMT) is a proposed model for the initiation of metastasis. During EMT cell-cell adhesion is reduced allowing cells to dissociate and invade. Of the EMT-associated transcription factors, ZEB1 uniquely promotes NSCLC disease progression. Here we apply two independent screens, BioID and an Epigenome shRNA dropout screen, to define ZEB1 interactors that are critical to metastatic NSCLC. We identify the NuRD complex as a ZEB1 co-repressor and the Rab22 GTPase-activating protein TBC1D2b as a ZEB1/NuRD complex target. We find that TBC1D2b suppresses E-cadherin internalization, thus hindering cancer cell invasion and metastasis.

[1] Department of Thoracic/Head and Neck Medical Oncology, The University of Texas MD Anderson Cancer Center, Houston, TX 77030, USA. [2] Department of Molecular and Cellular Oncology, The University of Texas MD Anderson Cancer Center, Houston, TX 77030, USA. [3] Department of Medical Biophysics, Princess Margaret Cancer Centre, University of Toronto, Toronto, ON M5S 1A1, Canada. [4] Department of Epigenetics and Molecular Carcinogenesis, The University of Texas MD Anderson Cancer Center, Houston, TX 77030, USA. [5] Department of Cancer Genomics, The University of Texas MD Anderson Cancer Center, Houston, TX 77030, USA. [6] Institute for Applied Cancer Science, The University of Texas MD Anderson Cancer Center, Houston, TX 77030, USA. [7] Department of Medicine and Dan L. Duncan Comprehensive Cancer Center, Baylor College of Medicine, Houston, TX, USA. *email: dlgibbon@mdanderson.org

Epithelial-to-mesenchymal transition (EMT) is the process by which epithelial cells lose their apical-basal polarity and concomitantly acquire a migratory phenotype[1,2]. Epithelial cells undergo EMT during normal embryonic development, allowing cells to migrate and differentiate, however EMT also facilitates tumor progression. The metastatic cascade represents a multi-step process and EMT empowers cancer cells to disseminate at the invasive fronts of tumors, intravasate, survive in the circulation, and extravasate into distant tissues. EMT is activated by multiple transcription factors (e.g., TWIST, SNAIL, SLUG, ZEB1/2), however in lung adenocarcinoma, zinc finger E-box binding homeobox 1 (ZEB1, δEF1, zfhx1a) expression is an early and pivotal event in pathogenesis; conferring metastatic properties, treatment resistance, and correlating with poor prognosis[3–7]. Hence, understanding how ZEB1 functions provides promise for innovative therapeutic strategies to improve lung cancer patient outcome, which remains the leading cause of cancer related death[8].

ZEB1 orchestrates EMT through repression of epithelial genes such as E-cadherin, a central component in adherens junctions, and the microRNA-200 family. ZEB1 represses transcription of target genes through the epigenetic regulation of promoter chromatin architecture[9]. Densely arranged heterochromatin regions restrict the access of the transcription machinery thereby limiting the expression of the underlying gene[10]. ZEB1 enhances heterochromatinization at target gene promoters by increasing H3K27 deacetylation and tri-methylation[9,11]. Class I and II HDAC inhibitors have been shown to have efficacy in restoring ZEB1 target gene expression, but the mechanism behind this association remains incomplete[11]. ZEB1 can interact with the C-terminal binding protein (CtBP) corepressors to aid in the recruitment of the CoREST complex in pancreatic tumors[12–14], however subsequent work has proposed that ZEB1 represses targets via CtBP-independent mechanisms in prostate cancer-suggesting that ZEB1 binding partners may be context specific[15].

To expand the repertoire of treatments for metastatic NSCLC, we utilized two independent screening approaches to identify ZEB1 interactors that are essential to cancer cell survival. Here we describe an interaction with the Nucleosome Remodeling and Deacetylase (NuRD) complex. The NuRD complex is one of four major chromatin remodeling complexes[16,17]. The catalytic core of NuRD complexes consists of the histone deacetylases (HDAC1/2) and the chromodomain helicase DNA-binding proteins (CHD4/Mi-2β and CHD3/Mi-2α), which also act as scaffolds for the other complex members. The current understanding of the biochemical and structural features of NuRD components suggests that combinatorial assembly of these factors confers functional specificity to the NuRD complex. For instance, CHD3 and CHD4 are found in mutually exclusive NuRD complexes with non-overlapping functions. Multiple biological functions are regulated through NuRD chromatin modification, and alterations in NuRD complex activity have been implicated in a broad range of human diseases, including cancer[18].

In this study, we find that ZEB1 recruits the NuRD complex in NSCLC and link this association to the repression of known ZEB1 target genes. Furthermore, we exploit the functional cooperation of ZEB1 and the NuRD complex to identify metastasis suppressors in NSCLC and establish the GTPase activating protein (GAP) TBC1D2b as a ZEB1/NuRD complex target gene. Mechanistically, we find that aberrant suppression of TBC1D2b contributes to the endocytosis and degradation of E-cadherin to promote EMT.

## Results

**BioID screen reveals ZEB1 interactome**. Previous studies have applied affinity purification-mass spectrometry (AP-MS) analysis to identify proteins in stable association with ZEB1[6,19]; however, a high-confidence ZEB1 protein interactome has yet to be established. To elucidate ZEB1 transcriptional co-regulators we applied the BioID screening method to identify ZEB1 interacting partners[20,21]. This technique harnesses an abortive *E. coli* biotin ligase (BirA-R118, denoted BirA*) fused to a protein of interest. BirA* can generate biotinoyl-AMP, but has lost the ability to interact with this intermediate. Highly reactive biotinoyl-AMP is thus released into the vicinity of the bait protein, and reacts with amine groups on nearby polypeptides. Biotinylated proteins can then be isolated with streptavidin and identified using mass spectrometry. In contrast to traditional AP-MS, this methodology allows for the elucidation of interactions that may be lost during stringent lysis and washing (Fig. 1a).

A FlagBirA* tag was fused in-frame to either the N-terminus or C-terminus of human ZEB1 and stably integrated into HEK293 Flp-In cells, under the control of a tetracycline-inducible promoter. Two isogenic pools were generated for each bait protein, representing biological replicates. We performed RT-qPCR and immunoblot to validate ZEB1 upregulation and to determine the effect on the expression of the known ZEB1 target gene, E-cadherin. Upon tetracycline induction, exogenous expression of ZEB1 consistently produced E-cadherin repression by both mRNA and protein expression (Fig. 1b, c and Supplementary Fig. 1a, b). To additionally assess the functionality of the FlagBirA*-tagged human ZEB1 proteins, we expressed the constructs in the murine 393P cell line (Supplementary Fig. 1c). Both tagged ZEB1 constructs significantly upregulated the migratory and invasive potential of the 393P cell line, confirming that the FlagBirA* tag did not hinder the biologic function of ZEB1 (Supplementary Fig. 1d). The expression of FlagBirA* protein alone (denoted as −) had no effect on E-cadherin expression levels or invasive potential in these assays.

Following validation of the biological activity of the N-terminal and C-terminal tagged ZEB1 proteins, cell pools were incubated with tetracycline, biotin and the proteasome inhibitor MG-132. Biotinylated proteins were isolated with streptavidin-sepharose beads, washed, and subjected to trypsin proteolysis. The released peptides were identified using nanoflow liquid chromatography–electrospray ionization–tandem mass spectrometry (nLC–ESI–MS/MS). Using cells expressing the FlagBirA* tag alone for comparison, the computational tool Significance Analysis of INTeractome[22] was used to assign confidence scores to individual protein-protein interactions with ZEB1. Proteins confidently identified with a Max SAINT score >0.8, identified with >2 unique spectra in both analyses, and with at least 2.5-fold greater peptide counts in the FlagBirA*-ZEB1 samples than in FlagBirA* samples, yielded a high-confidence list of 68 ZEB1 interacting proteins (Table 1, extended list found in Supplementary Data 1). Notably, BioID identified an association between ZEB1 and several HDAC1 and HDAC2 containing co-repressor complexes: Sin3, CoREST, and the NuRD complex. In fact, all core members of the NuRD complex were identified as the top-ranking hits in the BioID screen. Several members of the NuRD complex were previously identified as ZEB1 interactors (Supplementary Fig. 1e).

**Loss-of-function screen identifies vulnerabilities in NSCLC**. To ascertain the significance of these ZEB1 interactors as therapeutic targets in metastatic NSCLC we utilized a previously published in vivo shRNA drop out screen methodology specifically concentrated on epigenetic regulators[23]. ZEB1-mediated epigenetic dysregulation is documented in metastatic NSCLC and a variety of cancer types, implying a causal role in disease pathogenesis. To differentiate epigenetic vulnerabilities between metastatic and

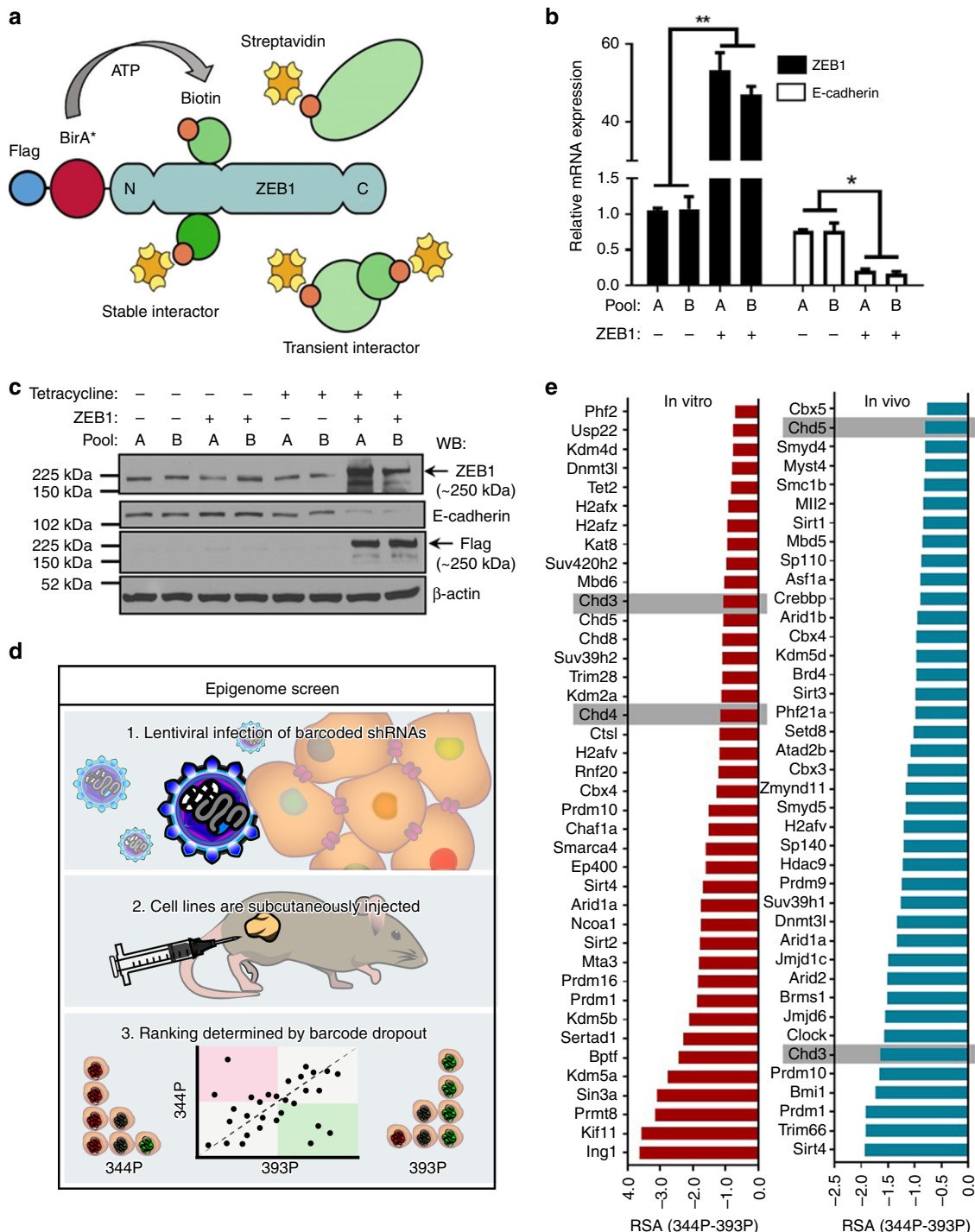

**Fig. 1** Biochemical and genetic screens reveal ZEB1 interacts with NuRD complex members. **a** Fusion of an *E. coli* abortive biotin ligase mutant (BirA*) to ZEB1 allows for biotinylation of transient or stable ZEB1 interacting proteins. Biotinylated proteins are captured by streptavidin conjugated sepharose beads and identified by mass spectrometry. Human *Zeb1* was cloned into the pcDNA5-FlagBirA*-FRT/TO vector and stably integrated in to HEK293 Flp-In cells. Subsequent to selection, cell lines were divided into two pools (denoted Pool A or B) and reflect biological replicates). Expression of the fusion protein repressed the established ZEB1 target, E-cadherin, as assessed by **b** qPCR and **c** immunoblot; all asterisks indicate statistical significance by *t*-test ($n \geq 3$, *$p \leq 0.05$); error bars represent standard error mean. **d** Depiction of the Epigenome short hairpin RNA (shRNA) dropout screen. Briefly, (1) an shRNA library consisting of 235 unique mouse or human epigenetic regulators was infected in to the murine Kras/p53 lung cancer cell lines, 393P and 344P. (2) Syngeneic 129/Sv mice were implanted with 400 cells/shRNA and monitored for four weeks; (3) Tumors (denoted In Vivo) and cell lines (denoted In Vitro) were sequenced to determine the barcoded shRNAs abundance and **e** rank was determined by differential analysis of 344P and 393P RSA score. Graphs represent top fifteenth percentile in in vitro and in vivo analyses and reveal hits with the most significant rank change between the mesenchymal 344P and the epithelial 393P cells

**Table 1 ZEB1 interactome**

| Gene Name | FlagBirA*-ZEB1 | | ZEB1-FlagBirA* | |
|---|---|---|---|---|
| | Total | SAINT | Total | SAINT |
| ZEB1 | 2006 | | 1214 | |
| CHD4 | 2273 | 1.00 | 629 | |
| MTA1 | 1700 | 1.00 | 324 | 1.00 |
| MTA2 | 1074 | 1.00 | 203 | 1.00 |
| HDAC1 | 773 | 1.00 | 142 | |
| GATAD2B | 673 | 1.00 | 129 | 1.00 |
| GATAD2A | 672 | 1.00 | 96 | 1.00 |
| HDAC2 | 639 | 1.00 | 106 | |
| RBBP7 | 638 | 1.00 | 131 | |
| RBBP4 | 603 | 1.00 | 116 | |
| CHD3 | 556 | 1.00 | 51 | |
| MTA3 | 526 | 1.00 | 71 | 1.00 |
| MBD3 | 510 | 1.00 | 74 | |
| CHD5 | 491 | 1.00 | 53 | |
| CHD8 | 394 | 1.00 | 28 | |
| ADNP | 269 | 1.00 | 47 | |
| ASUN | 229 | 1.00 | 11 | 0.93 |
| WDR82 | 173 | 1.00 | 24 | |
| IPO8 | 162 | 1.00 | 57 | 1.00 |
| MBD2 | 155 | 1.00 | 7 | |
| KDM1A | 127 | 1.00 | 84 | 1.00 |

High confidence FlagBirA*-ZEB1 interacting proteins. Mass spectrometry data was analyzed as described in the text. Proteins identified with a max SAINT score >0.8, proteins identified with >2 unique peptides and with spectral counts at least 2.5-fold higher in FlagBirA*-ZEB1 samples are shown

non-metastatic NSCLC, we enlisted two murine cell line models derived from the genetically engineered $Kras^{LA1/+}$;$p53^{R172H\Delta G/+}$ (KP) mice[24,25]. We have previously described the KP model to faithfully recapitulate features of metastatic lung cancer patients. Aimed at understanding the drivers of metastatic disease our group derived a panel of lung adenocarcinoma cell lines from the KP model. Subcutaneous injection of the KP cell lines into syngeneic mice established the cell line 393P to be an epithelial and non-metastatic phenotype, while the cell line 344P is mesenchymal and has metastatic ability[25].

A barcoded shRNA library, previously published by Carugo et. al., targeting 235 unique epigenetic regulators was infected into 393P and 344P cells[23]. Target regulators included subunits of various complexes that remodel nucleosomes, catalyze post-translational modifications, deposit histone variants and methylate DNA. To enhance the robustness of the screen and facilitate hit prioritization, the library was designed with ten unique shRNAs targeting each gene. To ensure adequate representation of the complexity of the deep-coverage shRNA library in mouse samples, mice were implanted with 400 cells/shRNA. Tumors were harvested at 150–200 mm³ and barcode abundance was quantified by sequencing (in vivo). The cell lines grown in vitro (in vitro) were also sequenced to potentially delineate genes that contribute to in vivo survival (Fig. 1d). To detect the top hits (or top scoring genes) emerging from the screens, we assigned p-values from RSA (Redundant shRNA Activity) scores (Supplementary Data 2). Comparison of the RSA values of 344P (mesenchymal) and 393P (epithelial) allowed for the identification of genes which are essential for growth of the mesenchymal metastatic model vs. the epithelial and non-metastatic model. The top 15% of most differentially regulated genes were compared to the BioID screen and yielded five genes: CHD3, CHD4, CHD5, CHD8, and MTA3. One of the most robust hits to emerge by both in vitro and in vivo screening was CHD4, which was also the most significant interactor in the BioID screen, suggesting the NuRD complex is central to the biology of mesenchymal NSCLC (Fig. 1e).

**ZEB1 interacts with the NuRD complex.** To validate the ZEB1-NuRD interaction we employed co-immunoprecipitation (co-IP) in the HEK293 Flp-In cells originally utilized to conduct the BioID screen. Upon FlagBirA*-ZEB1 immunoprecipitation we were able to co-IP HDAC1 and through immunoprecipitation of MTA1 we observed co-IP of ZEB1 (Fig. 2a, Supplementary Fig. 2a). Furthermore, using exogenously expressed GFP-conjugated ZEB1 in the 344SQ murine cell line (Supplementary Fig. 2b), we were also able to detect an interaction between ZEB1 and the NuRD complex members HDAC1 and MTA1 by co-IP (Fig. 2b). We next explored the endogenous interaction of ZEB1 with each member of the NuRD complex through the application of the proximity ligation assay (PLA). The PLA technique employs oligonucleotide labeled species-specific secondary antibodies, which when within close proximity (30–40 nm) allow the oligonucleotides to be ligated for amplification of the resulting circular DNA. The amplified signal may then be visualized by the use of fluorescently-labeled oligonucleotides and fluorescence microscopy. Addition of the ZEB1 or HDAC1 specific antibodies alone yielded only background levels of fluorescence in H157 cells. However, focal nuclear fluorescence signals were detected when cells were probed with both the ZEB1 and HDAC1 antibodies (Fig. 2c and Supplementary Fig. 3). Similar results were observed for HDAC2, CHD4, MTA1, MTA2, and MTA3 in the human and murine NSCLC cell lines H157, H1299, 344SQ, and 531LN2, providing additional evidence that ZEB1 interacts with the NuRD complex (Fig. 2d and Supplementary Fig. 3). Considering that the MTA family of proteins form mutually exclusive NuRD complexes, frequently with non-overlapping function, we found it interesting that PLA detected an interaction with all MTA members. Given that the CHD proteins also form exclusive complexes, we noted that ZEB1 preferentially forms a complex with CHD4/NuRD in NSCLC cell lines.

To further substantiate the observation that ZEB1 interacts with the NuRD complex, human H157 and murine 344SQ lung cancer cell lines were utilized to conduct gel chromatography. We determined the apparent molecular weight of ZEB1 by applying nuclear lysates to a Superose column. The eluate was collected in 60 sequential fractions of equal volume and analyzed by SDS-PAGE, followed by immunoblotting for ZEB1. Native ZEB1 from both H157 (Fig. 2e) and 344SQ (Fig. 2f) cells eluted with an apparent molecular mass much greater than that of the predicted mass (125 kDa). ZEB1 immunoreactivity was detected in chromatographic fractions from the Superose column in one distinct peak greater than 669 kDa. Significantly, the elution pattern of ZEB1 overlapped with that of the NuRD complex proteins HDAC1 and CHD4, further supporting the finding that ZEB1 interacts with the NuRD complex.

**The CHD4-containing NuRD complex is a ZEB1 co-repressor.** Previously, ZEB1 was found to recruit class I and II HDACs in pituitary organogenesis through the formation of a complex containing CtBP and the CoREST corepressors[26]. However, our discovery implicated the NuRD complex in ZEB1-mediated repression. In order to explore the significance of the physical association between ZEB1 and the NuRD complex, we analyzed established transcriptional targets of ZEB1 by chromatin IP (ChIP). Since the chromodomain helicase DNA binding proteins CHD3 and CHD4 participate in distinct forms of the NuRD complex and given our observation that ZEB1 interacts predominantly with CHD4 in NSCLC, we designated CHD4 as a surrogate for the NuRD complex. In these experiments, ChIP was performed in H1299 cells with antibodies against ZEB1, CHD4, and H3K27ac. This modification was selected following previous studies, which suggested the CHD4/NuRD complex specifically

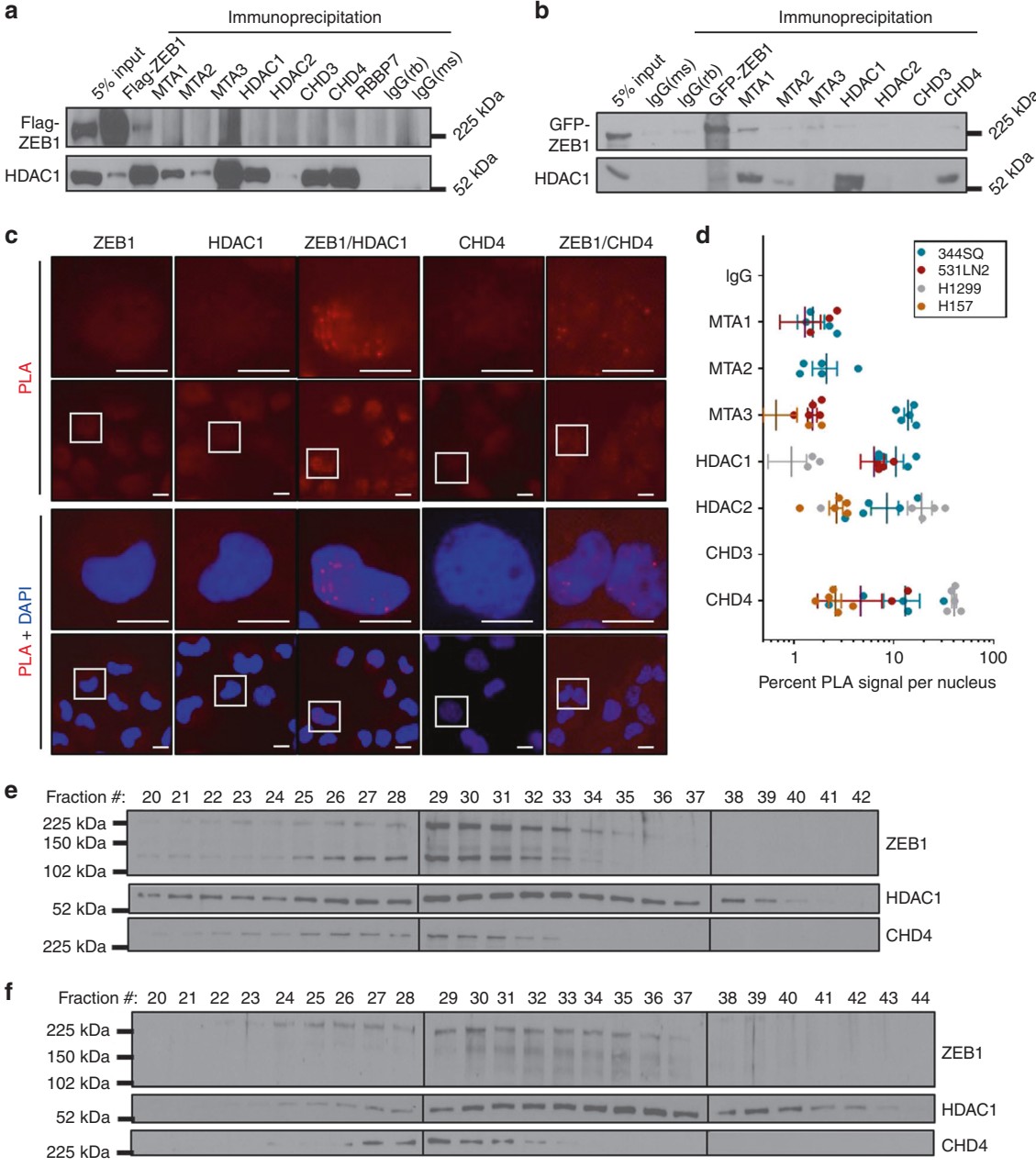

**Fig. 2** ZEB1 interacts with the NuRD complex. Co-immunoprecipitation of Flag-ZEB1 in HEK/293 Flp-In (**a**), or co-IP of GFP-ZEB1 in the murine cell line 344SQ (5% input indicates whole cell lysate) (**b**). Blots for HDAC1 and ZEB1 (Flag-tagged or GFP-tagged) confirm ZEB1 interacts with HDAC1 and MTA1. **c** Validation of ZEB1/NuRD complex interaction by proximity ligation assay (PLA). Human and murine NSCLC cell lines were fixed, permeabilized, blocked, and probed with species specific primary antibodies directed against ZEB1 and NuRD complex members. PLA was performed utilizing the Duolink In Situ Red Starter kit (Sigma). Representative PLA signal and nuclear staining (DAPI) in the human NSCLC cells H157 are shown. Red signal signifies interaction between ZEB1 and HDAC1. Scale bars represent 100 μm. **d** Quantification of mean PLA signal per cell in various murine and human cell lines; standard deviation n = 5. Results from gel filtration conducted in **e** 344SQ and **f** H157 showing ZEB1 elutes in fraction number 25–35, corresponding with a megadalton-sized complex. Blots of the NuRD complex members CHD4 and HDAC1 confirm that the NuRD complex members concurrently reside within fractions 25–35. Superose column standards: F47/48 = 669 kDa, F51 = 440 kDa, F58 = 158 kDa, F61 = 44 kDa

demethylates H3K27 to recruit the Polycomb Repressive Complex 2 (PRC2)[27]. We additionally depleted CHD4 to understand the significance of the NuRD complex to the regulation of the promoter activity of established ZEB1 genes. We selected miR-200c and SEMA3F as established ZEB1 target genes and included the CHD4/NuRD regulated gene, N-Myc, and SEMA3F intron 13 to be a negative control for ZEB1 binding[9,11,28]. Upon depletion of CHD4 we observed reduced CHD4 binding at all of the loci queried (Fig. 3a; Supplementary Fig. 4a). We detected ZEB1 and

CHD4 at the promoter of miR-200c and SEMA3F, but ZEB1 did not localize to the N-Myc promoter or SEMA3F intron 13 (Fig. 3a, b). We observed that ZEB1 binding to both the miR-200c and SEMA3F promoters was increased upon knockdown of CHD4. Yet, despite this increase in ZEB1 binding we observed an increase of H3K27ac at each of the CHD4 co-localized regions, suggesting that ZEB1 was not capable of repressing these genes in the absence of CHD4/NuRD recruitment (Fig. 3c). Treatment of H1299 with the class I/IV HDAC inhibitor, mocetinostat (1 μM

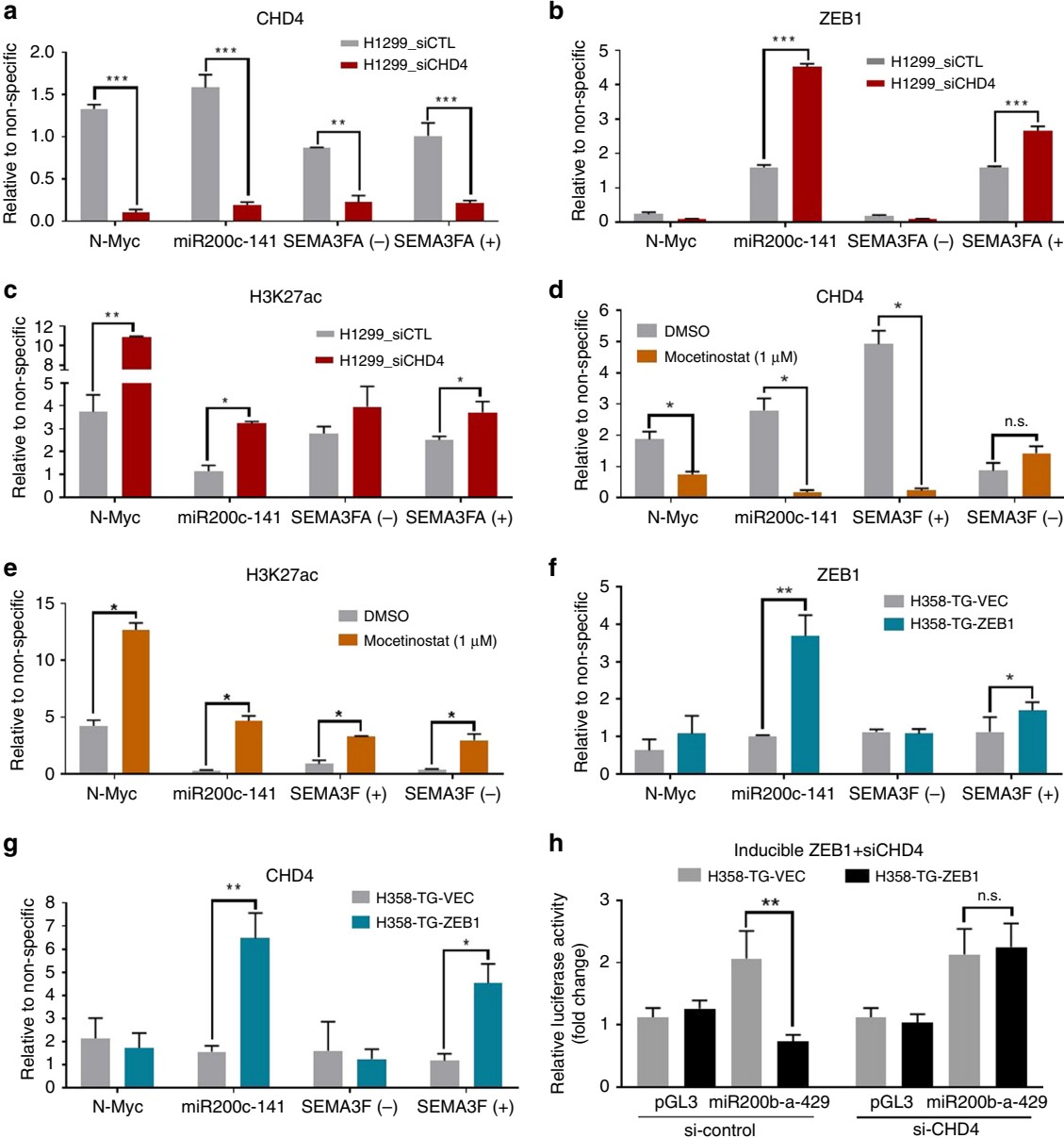

**Fig. 3** The CHD4/NuRD complex is a ZEB1 co-repressor. Chromatin immunoprecipitation (ChIP) for ZEB1, CHD4, H3K27ac, and control IgG was performed for known ZEB1 target genes (SEMA3F and miR-200c) and established CHD4/NuRD complex target (N-Myc) in the NSCLC cell line H1299. Primers corresponding to SEMA3F intron 13 (SEMA3F-neg) were included as a negative control for ZEB1 binding. When indicated, ChIP experiments were normalized to the non-specific IgG control. Graphs represent three experimental replicates; all asterisks indicate statistical significance by t-test ($n \geq 3$, *$p \leq 0.05$); error bars represent standard error mean. Transient knockdown of CHD4 by siRNA significantly reduced CHD4 binding at each of the queried promoters (**a**) but promoted ZEB1 binding (**b**). Despite increase ZEB1 binding, acetylation of H3K27 (**c**) increased at the miR-200c and SEMA3F promoter. **d** Relative CHD4 binding in the cell line H1299 after 24 h of treatment with the class I HDAC inhibitor, mocetinostat (1 μM). **e** Reduced CHD4 binding is observed upon treatment, corresponding with increased H3K27ac. **f, g** In the human NSCLC cell line H358, ZEB1 or GFP vector were overexpressed for 24 h prior to ChIP. ZEB1 and CHD4 concurrently bind to the promotor of miR-200 and SEMA3F, but not N-Myc. In addition, CHD4 binding to the promoter of miR-200c and SEMA3F occurs only upon ZEB1 overexpression. **h** Relative normalized luciferase activities from reporter constructs of either empty vector (pGL3), or a 321 base pair upstream fragment of the miR-200b-a-429 promoter[29]. Vectors were transfected in H358-GFP control or ZEB1-expressing cells, which were pre-transfected with siRNA control or siRNA targeting CHD4. Doxycycline induction was conducted prior to quantification of luciferase activity. Graphs represent normalization to control; all asterisks indicate statistical significance by t-test ($n \geq 3$, *$p \leq 0.05$); error bars represent standard error mean

for 24 h), similarly decreased CHD4 binding at N-Myc, miR200c and SEMA3F-positive, while concurrently increasing H3K27ac at all of the queried loci (Fig. 3d, e). In contrast to CHD4 knockdown, mocetinostat did reduce ZEB1 binding at the SEMA3F promoter, but did not affect binding to the miR-200c promoter (Supplementary Fig. 4b).

To determine whether ZEB1 can orchestrate CHD4/NuRD recruitment we expressed a doxycycline inducible GFP-ZEB1 in the human NSCLC cell line H358 (H358-GFP-ZEB1). Upon ZEB1 expression we observed a phenotypic EMT, which was confirmed by downregulation of E-cadherin (Supplementary Fig. 4c). We subsequently performed ChIP for CHD4 and ZEB1.

Consistent with the BioID and phenotypic data, we detected co-binding of ZEB1 and CHD4 at the miR-200c-141 and SEMA3F promoters only in the ZEB1 overexpressing cells, suggesting that ZEB1 enhances CHD4-NuRD binding at these sites (Fig. 3f, g). By contrast we found CHD4 binding at the N-Myc promoter was not influenced by ZEB1 overexpression and we did not find ZEB1 binding to the SEMA3F intronic region under either circumstance.

We also utilized the H358-GFP-ZEB1 cell line to ascertain the significance of CHD4 in the ZEB1-dependent repression of the miR200a-b-429 cluster. We performed a luciferase reporter assay, using the luciferase coding region cloned downstream of the miR-200a-b-429 promoter[29] and transfected into the H358_GFP or H358_GFP-ZEB1 cells prior to doxycycline induction for 24 h (Supplementary Fig. 4d). As expected, induction of ZEB1 expression decreased the luciferase reporter activity as compared to the GFP control (Fig. 3h). To determine whether ZEB1 was capable of repressing the miR-200a-b-429 promoter expression in the absence of CHD4, we transiently knocked down CHD4 prior to expression of ZEB1. In the ZEB1 expressing cells we observed a rescue of the luciferase activity upon CHD4 knockdown, suggesting that ZEB1 regulates miR-200 expression through a CHD4/NuRD complex (Fig. 3h). Consistent with these results, the endogenous gene expression of miR200b and miR141 was similarly restored upon CHD4 depletion (Supplementary Fig. 4d).

**Identifying targets of a ZEB1/NuRD complex**. We postulated that defining ZEB1/NuRD target genes may uncover regulatory pathways contributing to NSCLC invasion and metastasis. To delineate direct transcriptional effectors of this complex we utilized the ZEB1 and CHD4 ChIP-seq data provided by the ENCODE project (Encyclopedia of DNA Elements)[30] (Fig. 4a). Overlapping DNA sequences/gene promoters were considered potential targets of a ZEB1/CHD4/NuRD complex, and identified a total of 7231 different genomic locations. We further filtered this list by analysis of several mRNA datasets to identify targets that are inversely correlated with ZEB1 expression. Datasets included comparison of ZEB1 overexpression in the 393P murine cells, miR-200 overexpression in murine 344SQ cells, ZEB1 knockdown in human MDA-MB-231 breast cancer cells, and ZEB1 overexpression in 3T3-L1 pre-adipocytes[19,24,31]. 93 candidates were repressed by at least 50% upon ZEB1 overexpression in the 393P cell line and were significantly changed in any other one dataset. Our last criterion was the location of the binding site in relation to the distance from the transcription start site, as ZEB1 has been reported to preferentially bind gene proximal regions (−/+250 bp from the TSS)[32]. This yielded 37 genes, which we sought to further validate (Table 2; see Supplementary Data 3 for complete list). We determined the capacity of ZEB1 to repress candidate targets by comparing the mRNA expression of each gene upon constitutive ZEB1 overexpression in the 393P cell line (pcDNA_ZEB1; Fig. 4b and Supplementary Fig. 5a, c) or inducible expression of miR200a-b-429 in the 344SQ cell line, which should rescue the target gene expression (Fig. 4c and Supplementary Fig. 5b, d). The potential target genes were stratified by fold change. Genes that were repressed by 50% or more upon ZEB1 overexpression or increased by 2-fold upon ZEB1 suppression were considered candidate ZEB1/CHD4/NuRD target genes. This yielded five genes: KCNK1, TBC1D2a, EPS8L2, DNM1, and TBC1D2b. Additional analysis of the upstream promoter sequence of the candidate target genes also revealed at least one E-box, in particular 'CACCTG', a previously described ZEB1 binding motif[33].

We verified whether ZEB1 and CHD4 bind to the candidate target gene promoters by employing ChIP in H1299 cells

(Fig. 4d). Compared to the non-specific control (IgG), ZEB1 and CHD4 co-occupied the promoters of KCNK1, EPS8L2, TBC1D2a and TBC1D2b, four of the five genes queried. To validate ZEB1 and CHD4 binding to the target promoters, CHD4 was depleted by siRNA prior to ChIP with antibodies against ZEB1, CHD4, and H3K27ac. CHD4/NuRD depletion resulted in marked reduction of the recruitment of CHD4 to the promoter of EPS8L2, TBC1D2a, and TBC1D2b, but not KCNK1, suggesting non-specific binding to the KCNK1 promoter (Fig. 4e). No observable trend in ZEB1 binding was perceived upon CHD4 knockdown (Supplementary Fig. 5e), however H3K27 acetylation increased at the TBC1D2a, TBC1D2b, and EPS8L2 promoters, signifying these were indeed targets of a CHD4/NuRD complex (Fig. 4f). Treatment of H1299 with mocetinostat similarly decreased CHD4 and increased H3K27ac at all of the queried loci, while having variable effect on ZEB1 binding (Fig. 4g, h, Supplementary Fig. 5f). To delineate whether ZEB1 could enhance CHD4 binding to these two target promoters we again utilized the H358-GFP-ZEB1 cell line. Inducible ZEB1 expression produced a significant recruitment of CHD4 to the TBC1D2a, TBC1D2b and EPS8L2 promoters compared to vector control cells (Fig. 4i, j).

**TBC1D2b/Rab22 axis mediates E-cadherin endocytosis**. Preliminary experiments transiently overexpressing each of the three hits yielded from the analysis of the ENCODE ChIP-seq data (Supplementary Fig. 6a, b) suggested that all three genes significantly affect in vitro migration and invasive potential of murine NSCLC (Supplementary Fig. 6c); however, TBC1D2b posed an intriguing candidate. Previous work has revealed TBC1D2b as a GTPase activating protein (GAP) for Rab22 family members (Rab22 and Rab31)[34]. Both Rabs are described in sorting recycling endosomes; however, Rab22 was recently identified in the membrane trafficking of clathrin-independent endosomes. In addition, Rab22 has previously shown to be required for lung cancer cell migration and invasion[35]. To study the role of TBC1D2b in lung cancer metastasis we inducibly expressed TBC1D2b or GFP in the murine and human cell lines 344SQ, 531LN2 and H1299. These cell lines exhibited a robust upregulation of TBC1D2b mRNA (Supplementary Fig. 7a) and protein expression (Fig. 5a) upon doxycycline induction. This observation was further confirmed by immunofluorescent staining, which revealed TBC1D2b localization to the cytoplasm (Supplementary Fig. 7b). We also observed a physical association between TBC1D2b and Rab22 by co-IP, but could not detect an interaction with Rab31 (Fig. 5b). We next investigated the functional role of TBC1D2b in tumor cell migration and invasion. Overexpression significantly reduced Transwell migration and invasion, as well as migration in a wound closure assay (Fig. 5c and Supplementary Fig. 7c). Conversely, stable TBC1D2b knockdown increased both Transwell migration and invasion (Supplementary Fig. 7d, e). To determine whether this phenotype was due to the TBC1D2b/Rab22 association we expressed a GFP-conjugated human Rab22 in 393P, H358, and H1299. Again, upon doxycycline induction we witnessed upregulation of Rab22 mRNA and protein expression (Fig. 5d and Supplementary Fig. 7f). Overexpression significantly upregulated Transwell migration and invasion (Fig. 5e), a phenotype previously observed by other groups[32]. To determine if TBC1D2b could avert this invasive phenotype we performed a rescue experiment in which TBC1D2b was transiently overexpressed in GFP-Rab22-expressing cells. After 24 h of TBC1D2b expression doxycycline was utilized to induce expression of Rab22 for 12 h. Subsequently, cells were harvested to confirm expression (Fig. 5f) and to perform Transwell migration and invasion assays. The cell line 393P

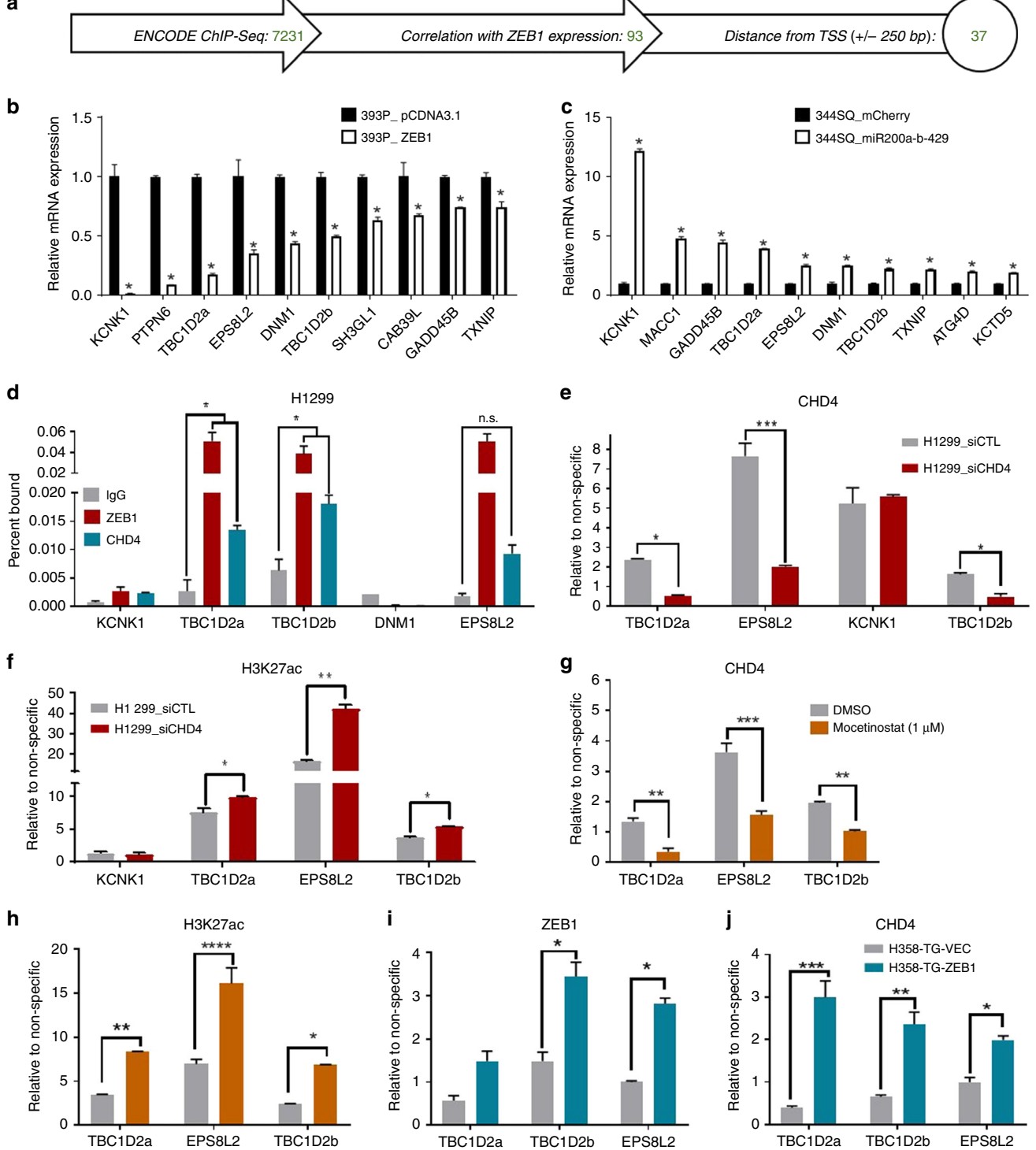

**Fig. 4** Defining targets of a ZEB1/CHD4/NuRD complex. **a** Flow chart depicts the criteria for choosing 37 candidate ZEB1/HDAC co-repressor complex target genes from ENCODE ChIP-sequencing data. Briefly, the ENCODE ChIP-seq data for ZEB1 and CHD4 was mined for potential co-binding. The list was refined by various mRNA datasets to identify targets which inversely correlate with ZEB1 expression. Lastly, ZEB1 binding ($-/+250$ bp) was applied as ZEB1 preferentially binds to this region. Validation of ZEB1/CHD4/NuRD targets was initially determined by comparison of mRNA in **b** 393P constitutively expressing ZEB1 and **c** 344SQ inducibly expressing miR200a-b-429; all asterisks indicate statistical significance by $t$-test ($n \geq 3$, *p $\leq$ 0.005); error bars represent standard error mean. **d** ChIP-qPCR in the human NSCLC cell line H1299 confirm that ZEB1 and CHD4 co-occupy the promoter of KCNK1, TBC1D2, TBC1D2b and EPS8L2, as compared to non-specific IgG control. **e** Transient depletion of CHD4 in H1299 decreases CHD4 binding, but increases H3K27 acetylation (**f**) at the promoters of TBC1D2a, TBC1D2b, and EPS8L2. **g** CHD4 and (**H**) H3K27ac ChIP was performed in the cell line H1299 after 24 h treatment with mocetinostat (1 μM). Relative CHD4 binding decreased upon treatment, while H3K27 acetylation increased. **j** ZEB1 overexpression in H358 significantly enhances both ZEB1 and **i** CHD4 binding to TBC1D2b and EPS8L2. Enrichment in ChIP experiments are relative to non-specific IgG control at each locus and is presented as the mean $+/-$ standard error mean from three independent experiments; all asterisks indicate statistical significance by $t$-test ($n \geq 3$, *p $\leq$ 0.05); error bars represent standard error mean

**Table 2 Candidate ZEB1/CHD4/NuRD targets**

| Gene | 393P_ZEB1/CTL | | 344SQ_miR429/CTL | | 3T3-L1_ZEB1/CTL | | MDA-MD-231 _shZEB1/ shCTL | |
| --- | --- | --- | --- | --- | --- | --- | --- | --- |
| | t-test | Fold | t-test | Fold | t-test | Fold | t-test | Fold |
| KCNK1 | 0.000 | 0.003 | 0.000 | 33.603 | 0.000 | 0.053 | 0.129 | 0.610 |
| TBC1D2 | 0.000 | 0.057 | 0.066 | 0.800 | 0.008 | 0.645 | 0.562 | 1.107 |
| MUC1 | 0.000 | 0.057 | 0.000 | 3.764 | 0.014 | 0.532 | 0.521 | 1.712 |
| C1orf210 | 0.005 | 0.062 | 0.016 | 1.659 | 0.000 | 0.095 | 0.038 | 0.348 |
| PTPN6 | 0.001 | 0.063 | 0.050 | 1.359 | 0.010 | 0.433 | 0.396 | 0.874 |
| MACC1 | 0.000 | 0.085 | 0.000 | 30.483 | 0.000 | 0.036 | 0.046 | 0.425 |
| ATG4D | 0.000 | 0.141 | 0.039 | 1.207 | 0.057 | 0.630 | 0.783 | 0.956 |
| IFNGR1 | 0.041 | 0.170 | 0.032 | 0.706 | 0.038 | 1.310 | 0.210 | 1.088 |
| EPS8L2 | 0.000 | 0.170 | 0.002 | 1.154 | 0.706 | 1.021 | 0.663 | 0.867 |
| GADD45B | 0.000 | 0.215 | 0.015 | 2.316 | 0.007 | 0.758 | 0.836 | 1.145 |
| DNM1 | 0.004 | 0.227 | 0.120 | 1.416 | 0.158 | 1.416 | 0.216 | 1.472 |
| FAM188A | 0.000 | 0.236 | 0.006 | 1.630 | 0.018 | 0.453 | 0.438 | 1.168 |
| GTPBP2 | 0.001 | 0.247 | 0.393 | 1.695 | 0.045 | 0.908 | 0.511 | 0.914 |
| TERF2IP | 0.032 | 0.282 | 0.144 | 3.603 | 0.001 | 1.910 | 0.992 | 0.996 |
| SCAF8 | 0.000 | 0.307 | 0.005 | 1.215 | 0.023 | 0.492 | 0.753 | 0.939 |
| PLLP | 0.006 | 0.318 | 0.010 | 0.549 | 0.004 | 1.674 | 0.442 | 1.306 |
| MPC1 | 0.001 | 0.320 | 0.159 | 0.746 | 0.030 | 1.393 | 0.410 | 0.849 |
| ANKRD17 | 0.086 | 0.348 | 0.005 | 1.511 | 0.060 | 0.716 | 0.255 | 0.752 |
| UBE2Q2 | 0.000 | 0.348 | 0.069 | 1.645 | 0.273 | 0.727 | 0.864 | 0.961 |
| TBC1D2B | 0.001 | 0.353 | 0.105 | 2.741 | 0.195 | 1.340 | 0.904 | 1.012 |
| EBI3 | 0.221 | 0.384 | 0.483 | 1.088 | 0.035 | 0.604 | 0.025 | 1.445 |
| FAM207A | 0.000 | 0.400 | 0.047 | 1.475 | 0.617 | 0.885 | 0.090 | 0.892 |
| CHKA | 0.002 | 0.424 | 0.004 | 2.358 | 0.372 | 0.802 | 0.194 | 1.437 |
| TXNIP | 0.000 | 0.428 | 0.549 | 0.598 | 0.583 | 1.096 | 0.621 | 1.095 |
| TTC21A | 0.368 | 0.435 | 0.122 | 0.539 | 0.132 | 0.390 | 0.028 | 1.247 |
| GTF2I | 0.125 | 0.436 | 0.042 | 1.696 | 0.230 | 0.802 | 0.226 | 0.727 |
| FGFR1OP2 | 0.001 | 0.440 | 0.039 | 0.818 | 0.001 | 0.656 | 0.918 | 1.023 |
| KCTD5 | 0.000 | 0.441 | 0.045 | 1.184 | 0.013 | 0.794 | 0.948 | 1.021 |
| IPMK | 0.000 | 0.464 | 0.016 | 1.652 | 0.400 | 0.954 | 0.123 | 1.171 |
| STXBP5 | 0.002 | 0.471 | 0.033 | 1.456 | 0.640 | 1.069 | 0.263 | 0.768 |
| MAN1A2 | 0.001 | 0.473 | 0.052 | 1.128 | 0.019 | 1.546 | 0.520 | 0.846 |
| DIAPH3 | 0.000 | 0.483 | 0.594 | 1.781 | 0.013 | 0.758 | 0.340 | 0.786 |
| SH3GL1 | 0.000 | 0.483 | 0.001 | 1.651 | 0.025 | 1.153 | 0.350 | 1.157 |
| CAB39L | 0.001 | 0.484 | 0.029 | 0.828 | 0.202 | 0.827 | 0.299 | 0.719 |
| CENPO | 0.000 | 0.485 | 0.286 | 1.461 | 0.371 | 0.868 | 0.748 | 0.972 |
| ZDHHC14 | 0.370 | 0.490 | 0.023 | 2.400 | 0.282 | 0.855 | 0.619 | 1.125 |
| CD164L2 | 0.476 | 0.499 | 0.010 | 2.065 | NA | NA | NA | NA |

Analysis of ENCODE ZEB1 and CHD4 ChIP-seq yielded 37 potential target genes with potential significance in NSCLC metastasis. Targets were correlated with four mRNA datasets and distance of binding site to transcription start site

has limited ability to migrate and thus we observed no significant difference upon TBC1D2b overexpression in the GFP control cells in either assay (Fig. 5g). However, in the Rab22 over-expressing cell line, TBC1D2b expression suppressed the migratory and invasive phenotype despite an increase in Rab22 transcriptional levels, suggesting that TBC1D2b hinders invasion through the regulation of Rab22 protein activity (Fig. 5g).

Interested in the significance of TBC1D2b expression in the context of EMT and metastasis, we also examined E-cadherin levels. Although there was no significant effect on E-cadherin transcription upon TBC1D2b manipulation until 24 h of induction (Supplementary Fig. 8a), an increase in E-cadherin protein levels was observed within 4 h of TBC1D2b expression (Fig. 6a) and conversely E-cadherin protein levels decreased upon Rab22 overexpression or TBC1D2b knockdown (Supplementary Fig. 8b, c). In addition, a faster migrating form of E-cadherin (97 kDa compared to the commonly observed 120 kDa) was detected by immunoblot upon TBC1D2b expression (Fig. 6a and Supplementary Fig. 6b). E-cadherin dephosphorylation is a precursor to internalization[36], and the accumulation of a lower molecular weight E-cadherin suggested that E-cadherin was no longer endocytosed. Treatment of 344SQ-TBC1D2b lysate with lambda

phosphatase produced a molecular weight shift comparable to 97 kDa (Supplementary Fig. 8d), advocating that TBC1D2b promotes the dephosphorylated E-cadherin. Upon induction of Rab22 overexpression in 393P we also observed degradation productions of E-cadherin at approximately 35 kDa, suggesting that Rab22 promoted E-cadherin degradation (Fig. 6b). We found that treatment with the lysosomal inhibitor hydroxychloroquine (HCQ) was able to prevent E-cadherin degradation, while a proteosomal inhibitor (MG-132) did not, signifying that Rab22 promotes E-cadherin lysosomal degradation (Fig. 6c). To determine if TBC1D2b plays a significant role in the regulation of E-cadherin degradation we co-expressed GFP-E-cadherin[37] and RFP-LAMP1 in the 344SQ_TBC1D2b knockdown cell lines to determine the fate of E-cadherin by live cell imaging. We observed the GFP-E-cadherin localized to the cell periphery in wildtype control cells, while in the TBC1D2b knockdown cells we observed co-localization of the GFP and RFP signals, suggesting that the absence of TBC1D2b directs E-cadherin to the lysosomal compartment for degradation (Fig. 6d). To determine whether TBC1D2b was regulating surface E-cadherin uptake we performed a biotin internalization assay (Fig. 6e, f). TBC1D2b was expressed for 6 h prior to labeling surface proteins with cleavable

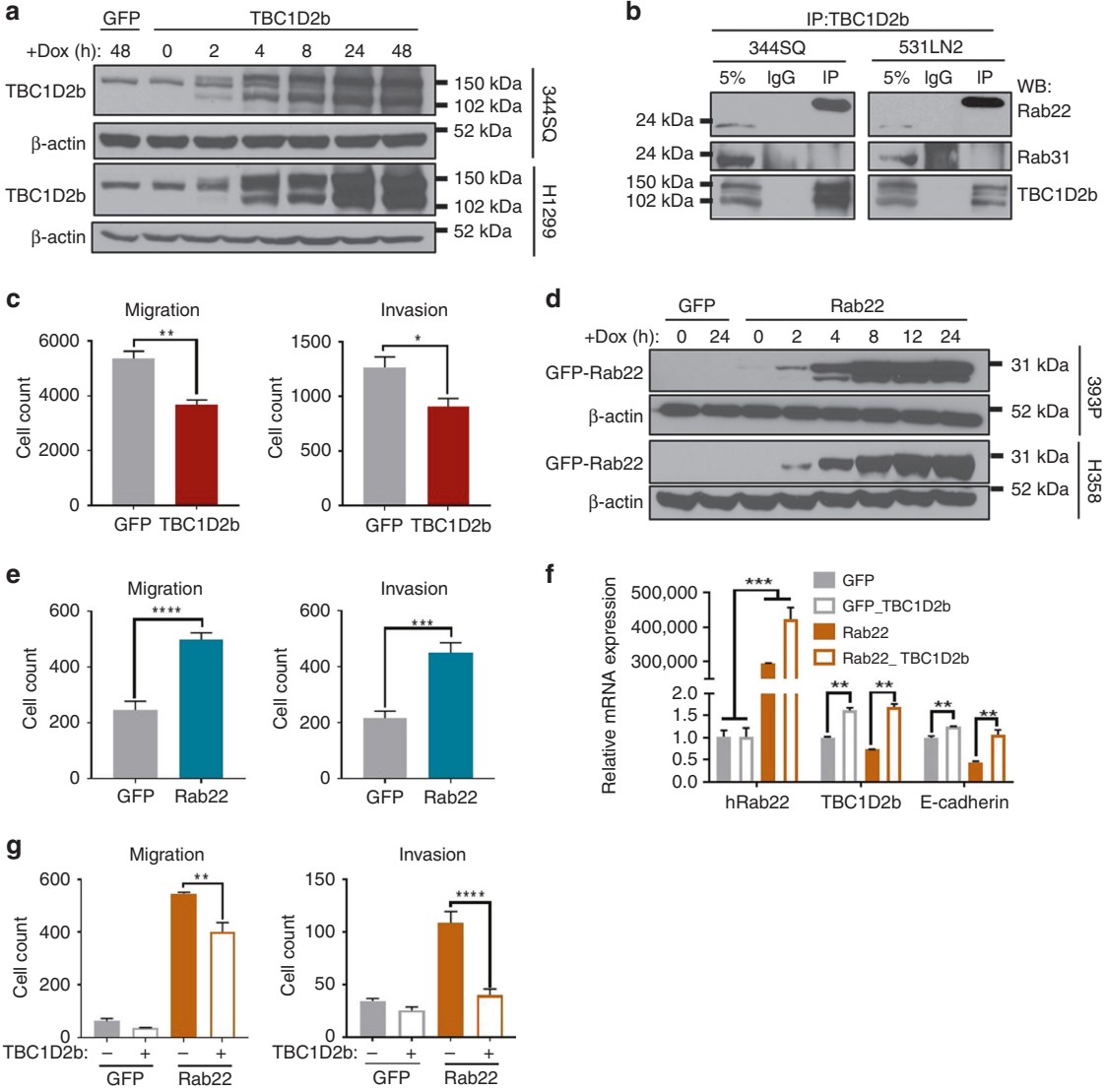

**Fig. 5** TBC1D2b hinders Rab22-mediated invasion. **a** Doxycycline-inducible murine TripZ-TBC1D2b construct was expressed in 344SQ and H1299 cell lines. Western blot demonstrates TBC1D2b expression over 48 h of time course induction. **b** Co-immunoprecipitation validates TBC1D2b interacts with Rab22 in 344SQ and 531LN2, however, no association was detected with Rab31 in these murine NSCLC cell lines (5% indicates whole cell lysate). **c** Quantification of Boyden chamber assay indicates 24 h of TBC1D2b overexpression diminishes cell migratory and invasive potential in 344SQ cell line; all asterisks indicate statistical significance by $t$-test ($n \geq 3$, $^*p \leq 0.05$); error bars represent standard error mean. **d** Western blot confirms overexpression of human GFP-Rab22 in 393 P and H358 cell lines. **e** 393P_GFP-Rab22 overexpressing cells exhibit increased Boyden chamber migration and invasion; all asterisks indicate statistical significance by $t$-test ($n \geq 3$, $^*p \leq 0.05$); error bars represent standard error mean. **f** pLenti-TBC1D2b construct was transiently expressed in 393P_GFP or 393P_GFP-Rab22 cell lines for 24 h prior to 24 h of doxycycline induction. mRNA confirms overexpression of TBC1D2b (48 h) and Rab22 overexpression (24 h). **g** In vitro transwell migration and invasion assay following TBC1D2b overexpression in 393P_GFP and 393P_GFP-Rab22 cell lines

biotin. Cells were then incubated at 37 °C for up to 1 h to allow protein internalization and surface biotin was cleaved to determine the relative amount of E-cadherin that was endocytosed. Compared with control cells, a reduced amount of biotinylated E-cadherin localized to the cytoplasm of TBC1D2b overexpressing 344SQ and 531LN2 murine lung cancer cells, suggesting that TBC1D2b regulates E-cadherin endocytic processing. Of note, the total amount of E-cadherin remained unchanged in both TBC1D2b and GFP control cell lines.

**TBC1D2b hinders NSCLC metastasis.** To determine the potency of TBC1D2b in metastasis in vivo, we implanted syngeneic mice subcutaneously with the TBC1D2b overexpressing or control

344SQ cells. Despite no difference in primary tumor growth, we observed a ~6-fold decrease in the number of distant lung metastatic nodules after 5 weeks (Fig. 7a). This was confirmed by haematoxylin and eosin staining of lung sections (Fig. 7c). Further analysis of TBC1D2b overexpressing tumors also confirmed an increase in TBC1D2b expression, which corresponded with an increase in E-cadherin mRNA and protein (Fig. 7b and Supplementary Fig. 9a). Conversely, we observed no significant difference in primary tumor size when comparing the growth of 344SQ tumors with constitutive TBC1D2b knockdown to the non-targeting control group (Fig. 7d, e and Supplementary Fig. 9b), but an increased number of metastatic lesions were detected, again supporting the hypothesis that TBC1D2b is a potent metastasis suppressor. Haematoxylin and eosin staining of lung

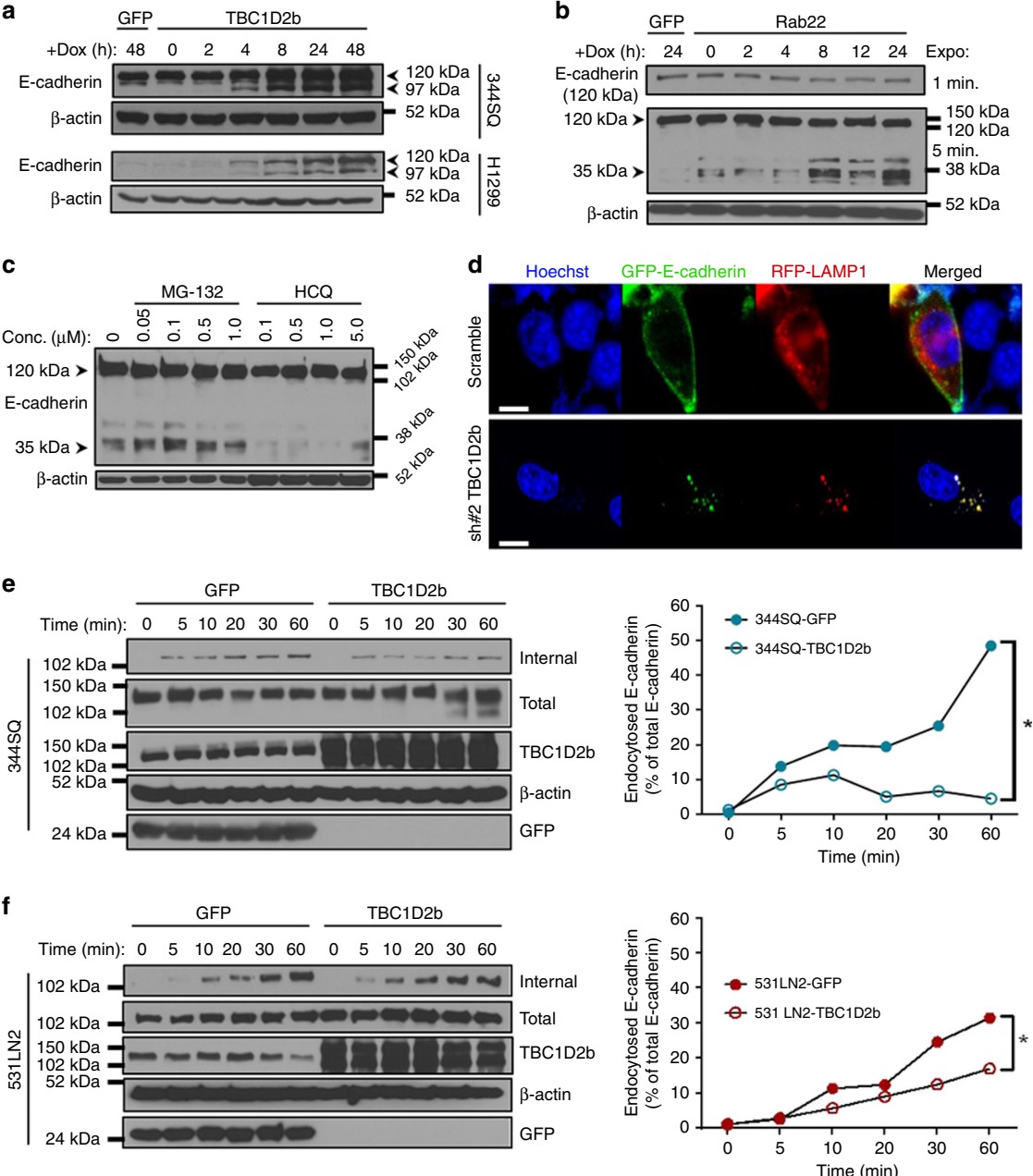

**Fig. 6** TBC1D2b/Rab22 axis regulates endocytosis of E-cadherin. **a** Immunoblot of E-cadherin in 344SQ and H1299 cells expressing TBC1D2b. Accumulation of a faster migrating band (~97 kDa) appears within ~4 h of TBC1D2b overexpression. Arrows designate different molecular weights of E-cadherin species. **b** Immunoblot of E-cadherin in 393P overexpressing human GFP-Rab22 over 24 h. E-cadherin (120 kDa) expression decreases upon Rab22 overexpression while a major degradation product (35 kDa) is observed and is dependent on Rab22 expression. **c** Treatment of 393P-GFP-Rab22 cell line with the lysosome inhibitor hydroxychloroquine (HCQ) or the proteasome inhibitor MG-132 at increasing concentrations for 24 h. **d** Upon TBC1D2b knockdown GFP-tagged E-cadherin co-localizes with RFP-LAMP1 lysosomes by live cell imaging in the 344SQ cell lines. Scale bars represent 10 μm. **e** E-cadherin internalization was assessed in the 344SQ-TBC1D2b and (**f**) 531LN2-TBC1D2b overexpressing cell lines through the biotin method. Quantification of western blot demonstrates that TBC1D2b suppresses E-cadherin internalization; all asterisks indicate statistical significance by $t$-test ($^*p \leq 0.05$)

sections confirmed the increased metastasis observed from gross examination (Fig. 7f).

Based on our overall findings we propose a model in which ZEB1 and CHD4/NuRD work in concert to repress TBC1D2b, as well as other targets such as the miR-200 family (Fig. 8). TBC1D2b downregulation results in an increase in Rab22 activation and in turn promotes E-cadherin internalization and degradation, enhancing in vivo metastasis.

## Discussion
The dysregulation of ZEB1 expression is associated with poor clinical prognosis in numerous epithelial cancers and notably drives EMT in lung cancer pathogenesis[3]. Previous work has shown that ZEB1 cofactors are critical to its function in tumorigenesis, metastasis, and therapy resistance[7]. Therefore in this paper we studied the molecular mechanisms governing ZEB1 function in metastatic NSCLC by applying two

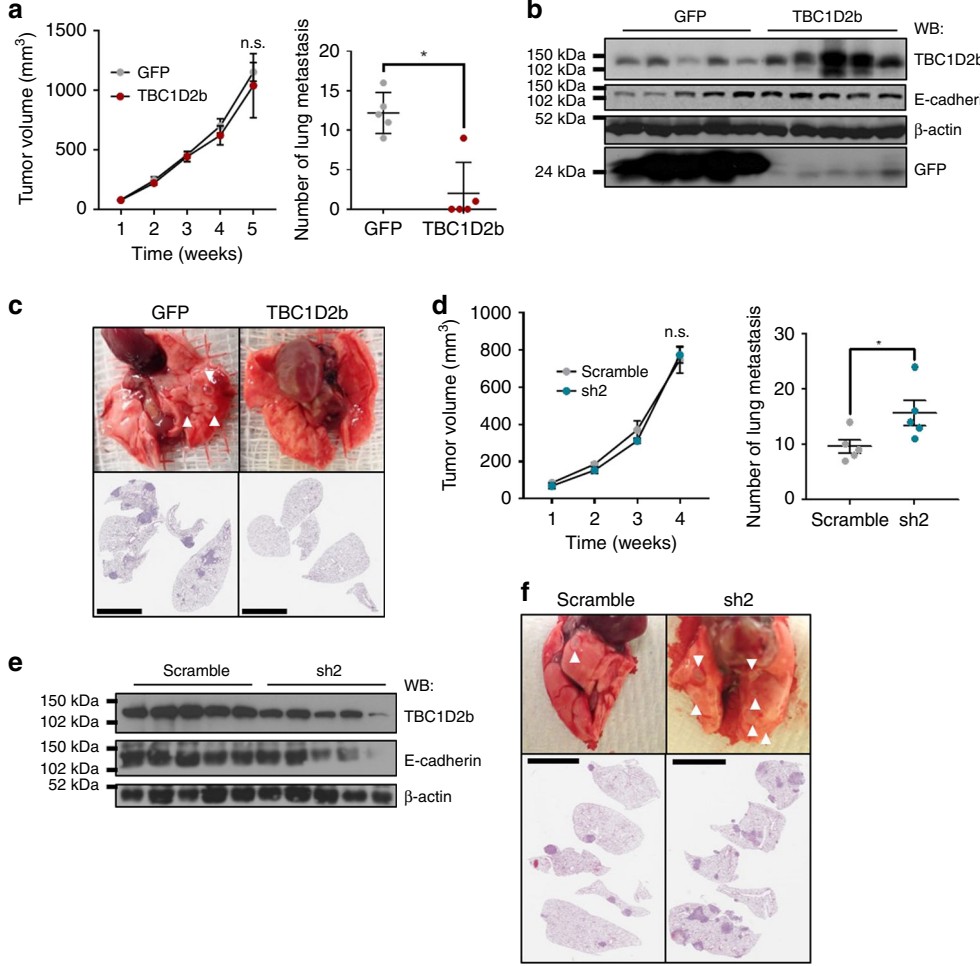

**Fig. 7** TBC1D2b overexpression in murine NSCLC cell lines inhibits metastatic potential. **a** 344SQ GFP control and TBC1D2b expressing cell lines were injected into the flank of 129/Sv mice; all asterisks indicate statistical significance by *t*-test ($n \geq 5$, *$p \leq 0.05$). The tumors were measured weekly and once tumors reached a maximum diameter of 1.5 cm, mice were euthanized. The number of lung metastases in the experimental mice versus the control were quantified manually. **b** Overexpression of TBC1D2b was confirmed by western blot and correlated with E-cadherin protein expression. **c** Representative lungs and respective H&E sections are shown. Arrows demonstrate metastatic lesions quantified. **d** 344SQ control or shRNA knockdown of TBC1D2b was performed similarly. Upon euthanasia, TBC1D2b (sh2) knockdown group was comparable in size to control (scramble) group; however, number of metastases was significantly greater in TBC1D2b knockdown group; all asterisks indicate statistical significance by *t*-test ($n \geq 3$, *$p \leq 0.05$). Scale bars represent 5 mm. **e** Western blot confirms knockdown and downregulation of E-cadherin upon depletion of TBC1D2b. **f** H&E staining of lung sections and representative lungs from knockdown experiment. Scale bars represents 5 mm

independent screens, the biochemical BioID screen[20] and an in vivo shRNA-mediated genetic loss-of-function screen[23]. This allowed us to study ZEB1 interactors that can be exploited therapeutically in metastatic NSCLC. Here we report that ZEB1 interacts with the NuRD complex and that chromodomain helicase family members are essential to the survival of metastatic NSCLC. Multiple studies have demonstrated that the NuRD complex associates with other transcriptional repressors, including FOG-1[38], and ZEB2 (Zfhx1b)[39–41]. Previous ZEB1 affinity purification studies have detected an interaction between ZEB1 and some NuRD complex members, although none followed up in studying the NuRD complex as a ZEB1 co-repressor[19,41]. To define the functional consequence of this interaction we considered established targets of ZEB1 and determined that not only does ZEB1 recruit CHD4/NuRD to target promoters, but in the case of miR-200, CHD4 is necessary to facilitate ZEB1-mediated repression. Aberrant DNA methylation of the miR-200c/141 promoter is closely linked to inappropriate silencing in cancer cells[42].

Interested in the identification of other ZEB1/CHD4/NuRD targets, we interrogated the ENCODE ChIP-seq data and through application of stringent criteria defined the paralogues TBC1D2a (Armus) and TBC1D2b as target genes. Rather than examining Armus, which was previously described as a Rac1 effector and a bona fide GAP for Rab7[43,44], we selected to characterize TBC1D2b (mKIAA1055) due to its association with lung cancer oncogenesis[45]. Earlier work established TBC1D2b as a Rab22 binding protein, including Rab22 (Rab22a) and Rab31 (Rab22b) [34]. TBC1D2b consists of two coiled-coil (CC) domains, which are flanked by an N-terminal pleckstrin homology (PH) domain and a C-terminal Tre-2/Bub2/Cdc16 (TBC) domain. Truncation mutants of each domain unexpectedly demonstrated that TBC1D2b has nominal GAP activity towards Rab22, more likely serving as a hub for the recruitment of other Rab22 GAPs[34]. Studies now place Rab22 at the level of recycling of clathrin-mediated endocytosis (CME) and clathrin-independent endocytosis cargo proteins[46,47]. There have been other reports of Rab22 acting at endocytic entry points, markedly in the endocytosis of

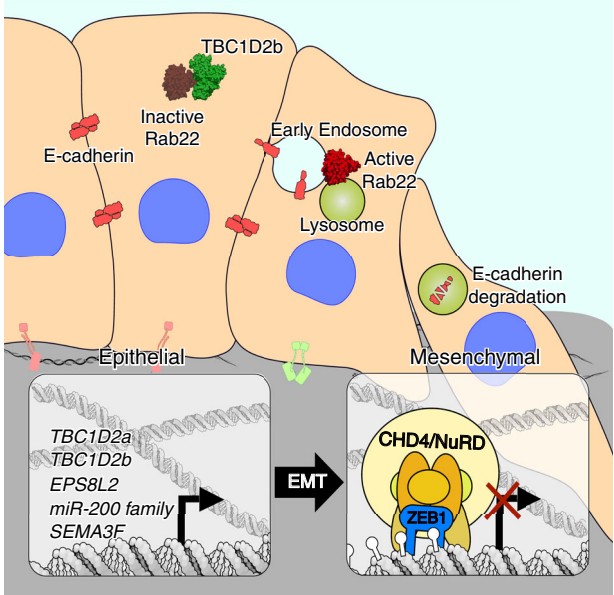

**Fig. 8** Working model. Proposed model demonstrating differential effects of TBC1D2b/Rab22 on E-cadherin internalization in epithelial and mesenchymal lung cancer cells due to ZEB1/NuRD-mediated regulation

the TrkA receptor[48] and in the early uptake of the bacterium *Borrelia*[49].

Initial overexpression of TBC1D2b led us to observe an upregulation and altered electrophoretic mobility of E-cadherin. It is well established that the cytoplasmic tail of E-cadherin is phosphorylated in the β-catenin binding region, increasing its affinity for β-catenin[36]. In addition, phosphodeficient E-cadherin mutants exhibit enhanced endocytosis and degradation through a lysosomal compartment. Together with the observed mobility shift we became intrigued in the role of TBC1D2b/Rab22 in the regulation of E-cadherin uptake. E-cadherin degradation is efficiently blocked by the expression of TBC1D2b, providing another important regulatory node for E-cadherin turnover and stability of cell-cell contacts. A wealth of reports have proven that EMT-associated transcription factors bind to the E-box within the E-cadherin promoter to suppress gene expression[50]; however, we provide evidence that ZEB1 can dually facilitate the down-regulation of surface E-cadherin by promoting excessive internalization and degradation. Our data also suggests that ZEB1 promotes hyper-activation of Rab22 and may regulate other junctional proteins, thereby disrupting tissue polarity and instigating a motile phenotype. Future studies will be required to determine whether ZEB1 influences the activity of Rab7 through regulation of TBC1D2a during normal development and tumor metastasis.

Concurrently, we determined that TBC1D2b is an inhibitor of invasion in vitro and metastasis in vivo. Given the physiological role of E-cadherin in cell–cell contacts, we propose that TBC1D2b likely stabilizes the cell junctions found in epithelial cells. In addition, we find that TBC1D2b rescues Rab22-mediated cell migration and invasion. Distinguishing the role of ZEB1-mediated Rab22 activation in the regulation of the endocytic pathway is an important area for future investigation. Furthermore, endocytosis entails selective packaging of cell surface proteins such as receptors that are frequently skewed in cancer cells[51]. Unveiling Rab22 cargo may yield invaluable tools for decoding therapeutic resistance in multiple epithelial tumors.

In conclusion, our results unveil the oncogenic function of the NuRD complex in NSCLC metastasis through physical association with and recruitment by ZEB1. The ZEB1/CHD4/NuRD complex is responsible for mediating repression of miR-200c/141 and TBC1D2b, a regulator of Rab22 and a potent suppressor of NSCLC invasion and metastasis. These findings demonstrate how EMT associated transcription factors regulate the degradation of E-cadherin protein and suggests that ZEB1/CHD4/NuRD can harness endocytosis to promote oncogenic signals. Therefore, this data validates the targeting of the ZEB1 axis for the treatment of metastatic lung cancer.

## Methods

**Cell culture**. Human lung cancer cell lines H157, H1299 and H358 were obtained from the National Cancer Institute (NCI-H series) or the Hamon Center for Therapeutic Oncology Research, University of Texas Southwestern Medical Center (HCC series). Cell lines from the KP mice were derived and maintained as previously described[25]. Cell line names depict the mouse number and site of derivation (e.g., 393P was derived from primary lung tumor). HEK/293 Flp-In T-Rex were provided by the Raught laboratory (University of Toronto) and were cultured in DMEM with 0.4% Hygromycin B. All other cell lines were cultured and passaged in RPMI 1640 supplemented with 10% fetal bovine serum (FBS) and incubated in 5% $CO_2$ at 37 °C. In separate experiments, cells were cultured for 4 h in the presence of hydroxychloroquine or MG-132.

**BioID**. Zeb1 was cloned from the pLenti-GIII-CMV-hZEB1-GFP-2A-Puro lentiviral vector (Applied Biological Materials Inc., LV362466, Accession No. BC112392) using PCR amplified with primers containing AscI and Not1 (N-terminus tag) or Kpn1 and NotI (C-terminus tag) restriction enzyme sites and cloned in to the pcDNA5 FRT/TO Flag-BirA$^{R118G}$ (pcDNA5 Flag-BirA*). BioID conducted as previously reported[52]. The N-terminus or C-terminus Flag-BirA tag vector control and Zeb1 were transfected into HEK/293 Flp-In along with pOG44 Flp-Recombinase expression vector using Lipofectamine® LTX Reagent with PLUS™ Reagent as per the manufacturer's instructions (Invitrogen, Cat. No. 15338100). Cell lines were cultured until colonies were ~3 mm in diameter at before divided into two pools. These were considered as a biological replicate and were processed independently.

Subsequently, all cell lines were cultured in four different conditions and harvested:

(1) DMEM
(2) DMEM + 5 µM MG-132
(3) DMEM + 1 µg/ml Tetracycline and 50 µM biotin
(4) DMEM + 1 µg/ml Tetracycline, 50 µM biotin, and 5 µM MG-132

Samples were snap-frozen and shipped on dry ice to the Raught laboratory for processing, mass spectrometry, and analysis.

**Epigenome screen**. A custom shRNA library targeting 235 epigenes (10 independent shRNAs/gene) was constructed by using chip-based oligonucleotide synthesis and cloned into the pRSI16 lentiviral vector (Cellecta) as a pool. Targeting sequences were designed using a proprietary algorithm (Cellecta). The oligo corresponding to each shRNA was synthesized with a unique molecular barcode (18 nucleotides) for measuring representation by NGS. Murine lung cancer cell lines (393P and 344P) were infected at a multiplicity-of-infection (MOI) of 0.3 with a pooled shRNA lentiviral library[23]. The GEMM-derived cells were transplanted at $10^6$ cells per mouse ensuring an in vivo representation of 400 cells/barcode. Illumina base calls were processed using CASAVA (v.1.8.2), and resulting reads were processed using our in-house pipeline. Raw FASTQ files are filtered for a 4-bp spacer (CGAA) starting at 18th base allowing for one mismatch, such that only reads amplified using above mentioned PCRs are used for further processing. We then extract 23–40 bp of the above reads for targeting libraries, and 1–18 bp for non-targeting library. These are further aligned using Bowtie (2.0.2) to their respective libraries (2.4 k mouse Epigenome and 12.5 k non-targeting library)[53]. Then use SAMtools to count the number of reads aligned to each barcode. Read counts are normalized for the amount of sequencing reads retrieved for each sample, using library size normalization. Fold change distribution was converted to percentiles, and biological replicates were collapsed for RSA analysis. The RSA logP-values and ranks are provided in Supplementary Data 2.

**Quantitative real-time PCR analysis**. Total RNA was isolated using TRIzol® Reagent (Life Technologies) according to the manufacturer's instructions and reverse transcribed using qScript™ Reagent (Quanta Biosciences). Analysis of mRNA levels was performed on a 7500 Fast Real-Time PCR System (Applied Biosystems) with SYBR® Green Real-Time PCR, using primers designed using the NIH primer design tool. The ribosomal housekeeping gene L32 was used as an internal control and data was analyzed with the 7500 Software v2.0.5 (Applied Biosystems). Student's *t*-test was performed for statistical significance (see Supplementary Table 1 for qPCR primer sequences).

**Immunoblot**. Protein estimation was conducted by use of the Pierce™ BCA Protein Assay Kit (Thermo Scientific, Cat. No. 23227). Samples were separated on SDS polyacrylamide gels and transferred onto a nitrocellulose membrane. The membranes were blocked using 5% nonfat dry milk and incubated in primary antibody overnight at 4 °C (uncropped images of blots are shown in Supplementary Fig. 10). Membranes were exposed using ECL (GE Healthcare) per the manufacturer's instructions.

**Immunofluorescence**. Cells were fixed with 4% paraformaldehyde, then permeabilized with 0.1% Triton X-100. The slides were incubated with the primary antibodies overnight at 4 °C. DAPI for the nuclear stain was contained in the mounting solution (see Supplementary Table 2 for complete antibody list). Images were acquired by confocal microscopy.

**Proximity ligation assay**. A total of $1 \times 10^5$ cells were seeded overnight onto glass coverslips. The next day, cells were fixed, permeabilized, blocked with BSA, and probed with primary antibodies. Cells were then treated with Duolink In Situ Red Starter Mouse/Rabbit kit (Sigma) as per manufacturer's instructions. Images were captured by confocal fluorescence microscope (Nikon).

**Immunoprecipitation**. Pull-down assays were performed using 500 μg of crude lysate incubated overnight with the primary antibody at 4 °C and gentle agitation. Protein A/G PLUS-Agarose Immunoprecipitation Reagent (SC-2003) was then introduced for 2 h. Antibody-antigen complexes were washed with phosphate-buffered saline (PBS) and Wash Buffer (50 mM Tris (pH 7.4), 150 mM NaCl, 0.1% NP-40), eluted with 1x RIPA buffer at 100 °C and separated by SDS-PAGE before Western blot analysis.

**RNA interference**. The human CHD4 siRNA SMARTpool was purchased from Dharmacon (L-009774-00-0005) and used at a final concentration of 25 nM. siRNA transfection was conducted using Lipofectamine RNAiMAX (Thermo Fisher Scientific). TRC Lentiviral pLKO.1 plasmid expressing scrambled control shRNA or murine TBC1D2b shRNA were purchased from Dharmacon (TRCN0000106070, TRCN0000106071, TRCN0000106072, TRCN0000106073, and TRCN0000106074). 344SQ and 531LN2 cells were virally infected as previously described and cultured in RPMI 1640 with 10% FBS and puromycin.

**Reporter assay**. For 3′UTR reporter assays cells were co-transfected with 500 ng of the reporter construct for 24 h and then assayed for luciferase activity after 24 h of doxycycline induction. For promoter-reporter assays cells were first pre-transfected for 24 h with siRNA followed by transfection for 24 h with 500 ng of the reporter constructs, then assayed for luciferase activity after 24 h of doxycycline induction. All reporter assays were performed using the Dual-Luciferase reporter assay system (Promega, Madison, WI).

**Chromatin immunoprecipitation**. Cells ($1 \times 10^7$) were fixed in 1.1% formaldehyde for 10 min at room temperature followed by quenching with 0.125 M glycine. Nuclei were isolated, and DNA was sheared by sonication to fragments of ∼300 bp. Chromatin was precipitated using the antibody ZEB1 (Santa Cruz) or anti-CHD4 antibody (Abcam). After reversal of cross-links, precipitated DNA was subjected to qPCR analysis using gene-specific primer pairs (see Supplementary Table 3 for ChIP primer sequences).

**Migration and invasion assay**. Cells were resuspended in serum-free media and seeded in a 24-well Transwell or Matrigel plate (BD Biosciences) at a concentration of $5 \times 10^4$ per well. RPMI supplemented with 10% FBS was added to the lower chamber and cells were allowed to migrate for 16 h in 5% $CO_2$ at 37 °C. Migrating cells were stained with 0.1% crystal violet. Non-migrating cells were removed using a cotton swab. Migrated cells were quantified based on five microscopic fields at a 4X magnification and results were represented as mean ± standard deviation and student's t-test was performed for statistical significance. Each assay was performed in triplicate.

**Biotin internalization assay**. Following doxycycline induction for 4 h, cells were washed with PBS containing 10 mM $CaCl_2$ and 1 mM $MgCl_2$ and incubated with 0.5 mg/ml EZ-Link NHS-SS Biotin (Pierce) for 30 min on ice, followed by washing with quenching reagent (15 mM glycine in PBS-$Ca^{2+}$-$Mg^{2+}$). Old media was added and cells were incubated at 37 °C for various time points to allow endocytosis. Biotinylated proteins at the plasma membrane were then stripped at 0 °C by glutathione treatment twice for 15 min (60 mM glutathione, 75 mM NaCl, 10 mM EDTA, 75 mM NaOH, and 1% BSA). Cells were lysed (1% Triton X-100, 150 mM NaCl, 50 mM Hepes pH 7.6 with protease inhibitors as above) and an aliquot was separated to measure total amount of E-cadherin. Biotinylated proteins (internalized) were recovered from lysates by immunoprecipitation with streptavidin

beads. The amount of internalized and total E-cadherin was quantified by Western blots.

**In vivo tumor and metastasis experiments**. All animal experiments were reviewed and approved by the Institutional Animal Care and Use Committee at The University of Texas M.D. Anderson Cancer Center. Cells were subcutaneously injected in the flanks of syngeneic 129/Sv mice of 8–10 weeks age and observed for tumor growth for a period of 5 weeks. Upon euthanasia, metastatic nodules on the surface of lung lobes were counted manually. Lung tissue was fixed in 10% formalin and then processed for sectioning followed by haematoxylin and eosin staining.

**Reporting summary**. Further information on research design is available in the Nature Research Reporting Summary linked to this article.

## Data availability

The source data underlying Fig. 1e, Table 1 and Table 2 are provided as Supplementary Data files 1–3. All the other data supporting the findings of this study are available within the article and its supplementary information files and from the corresponding author upon reasonable request. A reporting summary for this article is available as a Supplementary Information file.

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

## Acknowledgements

This work was supported by the NIH R37 CA214609-01A1, CPRIT-MIRA RP160652-P3, and Rexanna's Foundation for Fighting Lung Cancer to D.L.G., CPRIT multi-investigator award RP120713 C2 (C.J.C.) and NIH grant P30 CA125123 (C.J.C.). R.M. was supported by the Ruth L. Kirschstein National Research Service Award Individual Pre-doctoral Fellowship to Promote Diversity in Health-Related Research (NCI F31 CA232403-01) and Dr. John J. Kopchick Fellowship. D.P. was supported by a CPRIT Graduate Scholar Training Grant (RP140106). D.L.G. is an R. Lee Clark Fellow of the University of Texas MD Anderson Cancer Center, supported by the Jeane F Shelby Scholarship Fund. Work in the BR lab was supported by the Canada Foundation for Innovation and The Princess Margaret Cancer Foundation. The work was also supported by the generous philan-thropic contributions to The University of Texas MD Anderson Lung Cancer Moon Shots Program. We would like to thank Dr. Andrew Gladden for providing critical feedback and reagents, Laura Gibson for technical assistance and the UTMDACC Department of Veterinary Medicine Facility.

## Author contributions

R.Ma. designed and carried out experiments, analyzed data, and wrote the paper; E.C. and B.R. performed mass spectrometry and analyzed BioID data; M.C.B., S.T.K., D.H.P., S.A.S., K.A. and J.J.F. assisted in the technical support and acquisition of data; R.Mi., M.C.P., A.C., C.B., J.K. and T.H. performed the Epigenome Screen and analysis; F.C. and C.C. performed bioinformatic and biostatistics analysis; D.L.G. supervised all aspects of the project and the writing of the paper. All authors provided critical feedback and writing of the final paper.

## Competing interests

The authors declare no competing interests.
