## [Peer Review File · Nature Communications]

Reviewers' comments:

Reviewer #1 (Remarks to the Author):

In this manuscript, Manshoury et al. explore the mechanistic basis of ZEB1 mediated regulation of epithelial-mesenchymal transition (EMT) in non-small cell lung cancer (NSCLC). They first perform BioID screening in HEK-293T cells to examine ZEB1 interactors, and identify multiple components of the NuRD complex. In parallel they conducted an epigenetic focused shRNA screen in non-metastatic and metastatic murine Kras-p53 (KP) cell lines, and identified CHD4 as the top hit essential for metastasis. They then confirmed that tagged ZEB1 immunoprecipitated HDAC1 and other complex members in 293T and KP cells and used PLA and biochemical fractionation to demonstrate endogenous co-localization/co-elution in H1299 NSCLC or H157 HNSCC cells. They use H1299 cells to perform CHIP of multiple ZEB1 targets and observed that CHD4 depletion increased H3K27Ac, even at loci where ZEB1 binding was increased in the absence of CHD4/NuRD recruitment. They used KRAS mutant H358 epithelial NSCLC cells to induce EMT via ZEB1 over-expression, and confirmed CHD4 dependent regulation of miR200a-b-429. Through integrative analysis ZEB1/CHD4 CHIP-Seq data and mRNA regulation by ZEB1/ miR200a-b-429 they identified 5 candidate genes involved in downstream EMT regulation, including TBC1D2a/b. Finally, they link TBC1D2b mediated Rab22 regulation of E-cadherin to EMT/metastasis downstream of ZEB1, demonstrating that suppression of TBC1D2b promotes Rab22 mediated lysosomal recycling of E-cadherin. Nicely they confirm that TBC1D2b over-expression inhibits metastasis, whereas suppression of TBC1D2b promotes metastasis in the KP murine model.

In general this is a comprehensive set of work that nicely identifies additional mechanistic details downstream of ZEB1 during EMT. While the HDAC1/2 association is already known, the relationship with NuRD is novel, as is the post-transcriptional effect on E-cadherin via TBC1D2b-Rab22 regulation.

Specific comments

1. Figure 2 – the association of ZEB1 with HDAC1 is known. It would be nice to show the CHD4 blot below HDAC1 across all of the IPs in 2A/B, even if direct interaction with tagged ZEB1 was not observed. As the authors point out in 2B CHD4 was able to pull down GFP-ZEB1, and it is possible the interaction is indirect via HDAC1. Similarly, in 2C it would be important to show the ZEB1/CHD4 PLA images, since this is the novel interaction, not HDAC1.

2. Figure 4 – TBC1D2a is at least as strong a hit as TBC1D2b. A better rationale would be helpful to why the authors selected TBC1D2b beyond a potential association between genetic variants and lung cancer susceptibility. Did the authors at least try over-expressing TBC1D2a compared to TBC1D2b and see if it impacts migration and/or E-cadherin levels? Since Rab7 can also regulate E-

cadherin, as well as EGFR recycling, it would be interesting to know whether TBC1D2a contributes as well to these phenotypes, or whether TBC1D2b is indeed the dominant mediator.

3. Figure 7 - More generally, to extend the relevance of this finding to human NSCLC it would also be helpful to know if TBC1D2b expression tracks inversely with EMT signatures in lung TCGA data

4. Figure 8 – Clarity could be improved further. For example, in the left panel the up arrow is referring to upregulated ZEB1 target genes whereas in the right panel it says E cadherin degradation. It would be clearer to show on the right that the ZEB1/NuRD complex is repressing TBC1D2b (and the same other genes). Then in the cellular diagrams above explicitly label that Rab22 is sequestered in left panel and show Rab22 mediated lysosomal trafficking of E-cadherin in the right panel, using the E cadherin degradation label next to the lysosome.

Reviewer #2 (Remarks to the Author):

Previously, it has been reported that the transcriptional repressor ZEB1, a major inducer of an EMT, is also a critical promotor of malignant progression of non-small cell lung cancer (NSCLC). While the E-cadherin gene and the miRNA-200 family gene are the major targets of ZEB1-mediated transcriptional repression, here Manshouri and co-workers have substantially expanded on the molecular details of ZEB1's transcriptional repression and on the list of genes repressed by ZEB1. Intriguingly, they report that one of the target genes is TBC1D2b, a GAP of Rab22, and a critical player in the Rab22-mediated endocytosis and degradation of E-cadherin protein itself. The authors have employed an elegant combination of BioID affinity purification and mass spectroscopy to identify the interactors of ZEB1 and in combination with a cutting-edge shRNA drop-out screen and additional computational analysis they have identified the NuRD complex with CHD4 as one of the major components as one of the major functional interactors required for ZEB1's repressive activities. Subsequently, using ChIP and other approaches they have identified a list of potential direct ZEB1 target genes that also require CHD4-containing NuRD. Some of these genes, in particular TBC1D2b and its GAP activity target Rab22 have been functionally validated by demonstrating their activity in modulating the endocytosis and lysosomal degradation of E-cadherin and thus in affecting cell migration, invasion and metastasis.

The experiments have been thoughtfully designed and properly controlled, the results have been adequately interpreted and the conclusions drawn by the authors are in most parts convincingly supported by the experimental evidence.

While the report has a logical flow, some of the detailed results are not well connected. In particular, the role of ZEB1/NuRD as transcriptional repressor is funneled into the role of one target gene, while many others are not further pursued. Also, the process of an EMT is not extensively analyzed by including a larger number of EMT markers. It is assumed that the authors rely on the ample knowledge on the role of ZEB1 in the regulation of an EMT. However, in conjunction with the specific interaction of ZEB1 and NuRD it may be important whether this complex regulates all of an EMT or only part of it, even with a specific subset of CHD4-dependent target genes.

Of course, the regulation of E-cadherin protein levels and thus function by TBC1D2b and Rab22, main targets of ZEB1/NuRD like the E-cadherin gene itself, closes the circle and explains the critical role of this axis in malignant progression. Yet, it remains unknown whether the axis reported here represents the full ZEB1 function during malignant progression. In this context, would treatment with HDAC inhibitors, as demonstrated by the Brabletz laboratory to inhibit an ZEB1-driven EMT, also repress the TBC1D2b-Rab22 axis and cell migration, invasion and malignant progression in the experimental systems used here?

Specific comments:

Throughout, all figure legends are very scarce in experimental details and lack the information required to be able to follow the experiments and the results. They rather summarize the results.

Figure S1B: the total levels of ZEB1 protein is not much higher than the endogenous levels after Tet-induction, yet there is a repression of E-cadherin expression. Is the BirA-Flag-ZEB1 more active? Can the fusion construct functionally replace endogenous ZEB1, for example after knock-down?

Figure 1C and S1B: the specific bands for ZEB1 should be indicated.

Line 100: Suppl. Fig. 1C not 1B.

Figure 1E: in vitro vs. in vivo: no explanation is offered in the legend.

Figure 2E,F: what is the higher MW ZEB1 on the immunoblot? Is the SDS-PAGE not under reducing conditions?

The overlapping of HDAC1 and CHD4 by size fractionation does not prove any interaction. The results only shows that ZEB1 exists in a higher molecular weight complex. Together, these results do not really contribute critical data.

Figure 3F: Has the luciferase-reporter assay been performed in a transient transfection scenario? If yes, what does that mean for modulating histone acetylation by NuRD on a plasmid construct? Can this experimental setup mimic the endogenous gene situation?

Figure 4: Why has EPS8L2 not been followed up?

Figure 5B, legend: what is the 5% lane?

Figure 6B: the molecular weight shift of E-cadherin from phosphorylated to non-phosphorylated is not obvious. Controls before and after phosphatase treatment, or else, should be shown to indicate the size difference.

Figure 6D: HOECHST not HOESCHT.

Legend to Figure 6: D is missing and thus shifts E and misses F.

Reviewer #3 (Remarks to the Author):

The authors describe the relationship with ZEB1 and aberrant EMT initiating metastasis. I find this manuscript suitable for publication given some minor revisions/improvements.

In order to better define the experiments that contributed to elucidating these key interactions, the authors should address the following:

1) There should be a more complete description of where these targets comprising 235 epigenetic regulators were originally designed. Further, a description of how the 10 shRNAs/target were defined would be meaningful for a reader attempting a similar experiment.

2) For Fig 1B and 1C, the pool A and B are defined as biological replicates in the text of the manuscript. That should also be defined in the figure legend. Further, the error bars should be defined in the figure legend for 1B. How many technical replicates and what do the bars indicate?

3) The methods section for "Quantitative Real-Time PCR analysis" should be expanded to describe the methods for generating a cDNA library following the RNA isolation. Also, this section should be linked to a table in the supplemental section describing the sequence of the primers that were used.

4) The body of the text describes the use of a dual-luciferase reporter system. The authors should describe in the figure legend or the figure itself if the 'relative luciferase' is actually normalized firefly/renilla luciferase that is then set as a fraction to the control treatment to describe the change in activity.

Response to Reviewer's Comments:

Reviewer #1:

In this manuscript, Manshoury et al. explore the mechanistic basis of ZEB1 mediated regulation of epithelial-mesenchymal transition (EMT) in non-small cell lung cancer (NSCLC). They first perform BioID screening in HEK-293T cells to examine ZEB1 interactors, and identify multiple components of the NuRD complex. In parallel they conducted an epigenetic focused shRNA screen in non-metastatic and metastatic murine Kras-p53 (KP) cell lines, and identified CHD4 as the top hit essential for metastasis. They then confirmed that tagged ZEB1 immunoprecipitated HDAC1 and other complex members in 293T and KP cells and used PLA and biochemical fractionation to demonstrate endogenous co-localization/co-elution in H1299 NSCLC or H157 HNSCC cells. They use H1299 cells to perform ChIP of multiple ZEB1 targets and observed that CHD4 depletion increased H3K27Ac, even at loci where ZEB1 binding was increased in the absence of CHD4/NuRD recruitment. They used KRAS mutant H358 epithelial NSCLC cells to induce EMT via ZEB1 over-expression, and confirmed CHD4 dependent regulation of miR200a-b-429. Through integrative analysis ZEB1/CHD4 ChIP-Seq data and mRNA regulation by ZEB1/ miR200a-b-429 they identified 5 candidate genes involved in downstream EMT regulation, including TBC1D2a/b. Finally, they link TBC1D2b mediated Rab22 regulation of E-cadherin to EMT/metastasis downstream of ZEB1, demonstrating that suppression of TBC1D2b promotes Rab22 mediated lysosomal recycling of E-cadherin. Nicely they confirm that TBC1D2b over-expression inhibits metastasis, whereas suppression of TBC1D2b promotes metastasis in the KP murine model.

In general this is a comprehensive set of work that nicely identifies additional mechanistic details downstream of ZEB1 during EMT. While the HDAC1/2 association is already known, the relationship with NuRD is novel, as is the post-transcriptional effect on E-cadherin via TBC1D2b-Rab22 regulation.

1. Figure 2 – the association of ZEB1 with HDAC1 is known. It would be nice to show the CHD4 blot below HDAC1 across all of the IPs in 2A/B, even if direct interaction with tagged ZEB1 was not observed. As the authors point out in 2B CHD4 was able to pull down GFP-ZEB1, and it is possible the interaction is indirect via HDAC1. Similarly, in 2C it would be important to show the ZEB1/CHD4 PLA images, since this is the novel interaction, not HDAC1.

We appreciate these suggestions and have included both in the revised manuscript. Immunoprecipitation of MTA2, MTA3, HDAC2, CHD3 and CHD4 suggested that each of these components do not physically associate with ZEB1, which we have included in Supplemental Fig. 2B and 2C. Due to the stringent conditions of this technique, we consequently performed PLA. Images for CHD4/ZEB1 PLA have been included in the revised Fig. 2C and Suppl. Fig. 3 to highlight this novel interaction.

2. Figure 4 – TBC1D2a is at least as strong a hit as TBC1D2b. A better rationale would be helpful to why the authors selected TBC1D2b beyond a potential association between genetic variants and lung cancer susceptibility. Did the authors at least try over-expressing TBC1D2a compared to TBC1D2b and see if it impacts migration and/or E-cadherin levels? Since Rab7 can also regulate E-cadherin, as well as EGFR recycling, it would be interesting to know whether TBC1D2a contributes as well to these phenotypes, or whether TBC1D2b is indeed the dominant mediator.

We thank the reviewer for this thoughtful suggestion. As ZEB1 overexpression failed to significantly enhance ZEB1 recruitment to the TBC1D2a promoter (Fig. 4I), we chose to concentrate the current study on TBC1D2b. Furthermore, the TBC1D2b/Rab22 axis posed an interesting candidate due to the association of Rab22 with lung cancer progression and invasive phenotype (Zhou et al. 2017). Additionally, we have also found that TBC1D2b can physically associate with Rab7 (data not shown), which we intend to explore in future work.

To address the Reviewer's question we have expressed TBC1D2a, TBC1D2b and EPS8L2 in the murine cell line 344SQ to test the phenotypic effects, as shown in the revised Suppl. Fig. 6A. We confirmed overexpression by comparison of mRNA and protein to the control vector expressing cell lines. We also performed transwell assays to determine the effect of each gene on the migration and invasion phenotype

of 344SQ (Suppl. Fig. 6C). We found that all three genes consistently downregulated *in vitro* migration and invasion.

However, we also probed for E-cadherin protein expression upon overexpression of each of the three genes. Again, we observed TBC1D2b overexpression significantly upregulated E-cadherin, while TBC1D2a and EPS8L2 did not. These data suggest that TBC1D2b is indeed the dominant mediator (out of the three) of E-cadherin internalization. We agree with the Reviewer that the other two genes may have roles in migration or invasion that will need additional study in the future, but are beyond the scope of this current manuscript.

3. Figure 7 - More generally, to extend the relevance of this finding to human NSCLC it would also be helpful to know if TBC1D2b expression tracks inversely with EMT signatures in lung TCGA data.

As suggested by the reviewer, we queried the TCGA lung adenocarcinoma dataset (TCGA-Provisional as queried using <http://www.cbioportal.org/>) to determine whether TBC1D2b expression could be associated with a previously published EMT signature or as a prognostic indicator of disease outcome (data not shown). We found that patients with low TBC1D2b expression did not correlate significantly with disease free survival or with the previously published EMT score (Byers et al. 2013). This may be due to the molecular heterogeneity across the samples in the lung cancer dataset or potentially related the early-stage nature of this dataset, since all of the samples were acquired from patients who had localized, operable disease.

4. Figure 8 – Clarity could be improved further. For example, in the left panel the up arrow is referring to upregulated ZEB1 target genes whereas in the right panel it says E cadherin degradation. It would be clearer to show on the right that the ZEB1/NuRD complex is repressing TBC1D2b (and the same other genes). Then in the cellular diagrams above explicitly label that Rab22 is sequestered in left panel and show Rab22 mediated lysosomal trafficking of E-cadherin in the right panel, using the E cadherin degradation label next to the lysosome.

We have made the changes suggested by the Reviewer in the revised manuscript and hope that these have improved the clarity of the model.

Reviewer #2:

Previously, it has been reported that the transcriptional repressor ZEB1, a major inducer of an EMT, is also a critical promotor of malignant progression of non-small cell lung cancer (NSCLC). While the E-cadherin gene and the miRNA-200 family gene are the major targets of ZEB1-mediated transcriptional repression, here Manshouri and co-workers have substantially expanded on the molecular details of ZEB1's transcriptional repression and on the list of genes repressed by ZEB1. Intriguingly, they report that one of the target genes is TBC1D2b, a GAP of Rab22, and a critical player in the Rab22-mediated endocytosis and degradation of E-cadherin protein itself. The authors have employed an elegant combination of BioID affinity purification and mass spectroscopy to identify the interactors of ZEB1 and in combination with a cutting-edge shRNA drop-out screen and additional computational analysis they have identified the NuRD complex with CHD4 as one of the major components as one of the major functional interactors required for ZEB1's repressive activities. Subsequently, using ChIP and other approaches they have identified a list of potential direct ZEB1 target genes that also require CHD4-containing NuRD. Some of these genes, in particular TBC1D2b and its GAP activity target Rab22 have been functionally validated by demonstrating their activity in modulating the endocytosis and lysosomal degradation of E-cadherin and thus in affecting cell migration, invasion and metastasis.

The experiments have been thoughtfully designed and properly controlled, the results have been adequately interpreted and the conclusions drawn by the authors are in most parts convincingly supported by the experimental evidence.

While the report has a logical flow, some of the detailed results are not well connected. In particular, the role of ZEB1/NuRD as transcriptional repressor is funneled into the role of one target gene, while many others are not further pursued. Also, the process of an EMT is not extensively analyzed by including a larger number of EMT markers. It is assumed that the authors rely on the ample knowledge on the role of ZEB1 in the regulation of an EMT. However, in conjunction with the specific interaction of ZEB1 and NuRD it may be important whether this complex regulates all of an EMT or only part of it, even with a specific subset of CHD4-dependent target genes.

Of course, the regulation of E-cadherin protein levels and thus function by TBC1D2b and Rab22, main targets of ZEB1/NuRD like the E-cadherin gene itself, closes the circle and explains the critical role of this axis in malignant progression. Yet, it remains unknown whether the axis reported here represents the full ZEB1 function during malignant progression. In this context, would treatment with HDAC inhibitors, as demonstrated by the Brabletz laboratory to inhibit a ZEB1-driven EMT, also repress the TBC1D2b-Rab22 axis and cell migration, invasion and malignant progression in the experimental systems used here?

1. Of course, the regulation of E-cadherin protein levels and thus function by TBC1D2b and Rab22, main targets of ZEB1/NuRD like the E-cadherin gene itself, closes the circle and explains the critical role of this axis in malignant progression. Yet, it remains unknown whether the axis reported here represents the full ZEB1 function during malignant progression. In this context, would treatment with HDAC inhibitors, as demonstrated by the Brabletz laboratory to inhibit a ZEB1-driven EMT, also repress the TBC1D2b-Rab22 axis and cell migration, invasion and malignant progression in the experimental systems used here?

We thank the Reviewer for recommending this insightful experiment, which is now included in the revised version. In agreement with the work of Meidhof et al. 2015, our group has recently published that mocetinostat reverts the EMT phenotype of metastatic NSCLC through restoration of the miR-200 family expression (Peng et al. 2019). To determine whether class I HDAC inhibitors can also regulate the ZEB1/NuRD target genes identified herein we treated the cell line H1299 with mocetinostat for 24 h. Subsequently we performed ChIP for ZEB1, CHD4 and H3K27ac to determine the binding at the miR200c-141, N-Myc, SEMA3Fa, EPS8L2, TBC1D2a, and TBC1D2b promoters. ZEB1 bound to the promoters of miR-200c-141, TBC1D2a, and EPS8L2, irrespective of treatment (Suppl. Fig. 4D and 5F). However, CHD4 binding (Fig. 3D and 4G) was down-regulated at each of the target loci, which further correlated with up-regulation of H3K27ac (Fig. 3F and 4H). These results suggest that HDAC inhibitors can also regulate the TBC1D2b/Rab22 axis.

2. Throughout, all figure legends are very scarce in experimental details and lack the information required to be able to follow the experiments and the results. They rather summarize the results.

We thank the Reviewer for this comment and have corrected the figure legends to provide additional experimental details.

3. Figure S1B: the total levels of ZEB1 protein is not much higher than the endogenous levels after Tet-induction, yet there is a repression of E-cadherin expression. Is the BirA-Flag-ZEB1 more active? Can the fusion construct functionally replace endogenous ZEB1, for example after knock-down?

The reviewer is correct in this observation and we sincerely thank them for catching this discrepancy between the mRNA and protein expression of the ZEB1-flagBirA* construct (C-terminus tag). After examination, we noted that the mRNA expression shown in Suppl. Fig. 1A was a duplication of the data for the flagBirA*-ZEB1 mRNA expression (N-terminus tag construct; Fig.1B). This mistake has been corrected in the manuscript. The expression of the C-terminus tagged ZEB1 is approximately 11-fold higher, which more appropriately corresponds with the protein expression in Suppl. Fig.1B.

To confirm the flagBirA* fusion constructs can functionally replace endogenous ZEB1 to drive EMT, endogenous mZEB1 (murine ZEB1) was transiently knocked down in the murine cell line 393P. Following knockdown, each cell line was transfected with flagBirA*, flagBirA*-hZEB1 or hZEB1-flagBirA* (human ZEB1). Exogenous overexpression of the fusion hZEB1 constructs consistently promoted an increase of the invasive phenotype in the presence or the absence of endogenous mZEB1 (Suppl. Fig. 1C and 1D), confirming that both of the fusion constructs can indeed functionally replace endogenous ZEB1.

4. Figure 1C and S1B: the specific bands for ZEB1 should be indicated.

An arrow has been added to indicate the FlagBirA*-ZEB1 in Fig. 1C and Suppl. Fig. 1B of the revised manuscript.

5. Line 100: Suppl. Fig. 1C not 1B.

We have corrected this mistake in the revised manuscript.

6. Figure 1E: in vitro vs. in vivo: no explanation is offered in the legend.

The following sentence has been included in the figure legend to differentiate the *in vivo* and *in vitro* screen results if Figure 4D: "Briefly, (1) an shRNA library consisting of 235 unique mouse or human epigenetic regulators was infected in to the murine Kras/p53 lung cancer cell lines, 393P and 344P. (2) Syngeneic 129/Sv mice were implanted with 400 cells/shRNA and monitored for four weeks; (3) Tumors (denoted "In Vivo") and cell lines (denoted "In Vitro") were sequenced to determine the barcoded shRNAs abundance and (E) rank was determined by differential analysis of 344P and 393P RSA score. Graphs represent top fifteenth percentile in *in vitro* and *in vivo* analyses and reveal hits with the most significant rank change between the mesenchymal 344P and the epithelial 393P cells."

7. Figure 2E,F: what is the higher MW ZEB1 on the immunoblot? Is the SDS-PAGE not under reducing conditions? The overlapping of HDAC1 and CHD4 by size fractionation does not prove any interaction. The results only shows that ZEB1 exists in a higher molecular weight complex. Together, these results do not really contribute critical data.

The reviewer's observation that the ZEB1 molecular weight on the reducing SDS-PAGE is larger than predicted is currently being addressed and is beyond the scope of the current study. Although ZEB1 has a predicted molecular weight of 125 kDa, multiple groups have reported discrepancies in the observed molecular weight (approximately 190-220 kDa), including Zhang et al. 2014. Our work has confirmed that the higher molecular weight band is indeed ZEB1, as evaluated by mass spectrometry (data not shown), and we are currently working to understand the basis of this observation.

We acknowledge that size fractionation does not prove direct association of ZEB1 with HDAC1 or CHD4. However, these results taken along with the co-IP and PLA results do suggest that ZEB1 is most prevalently found in large molecular weight complexes containing HDAC1 and CHD4, consistent with its involvement in the NuRD complex.

8. Figure 3F: Has the luciferase-reporter assay been performed in a transient transfection scenario? If yes, what does that mean for modulating histone acetylation by NuRD on a plasmid construct? Can this experimental setup mimic the endogenous gene situation?

Based upon inclusion of additional data this figure is now Figure 3H.

To evaluate whether the experimental setup of the luciferase assay mimics the endogenous gene expression we examined the expression of the endogenous miR-200 family members, miR-200b and miR-141 upon the absence of CHD4 and overexpression of ZEB1 in the cell line H358 (Suppl. Fig. 4C). Analogous to the results of the luciferase reporter assay in Fig. 3H, we confirmed that the expression of these two miR-200 family members are rescued upon CHD4 knockdown.

9. Figure 4: Why has EPS8L2 not been followed up?

As CHD4 did not significantly bind to the EPS8L2 promoter in experiment Fig. 4D, it did not meet the criteria for a ZEB1/NuRD target gene. However, we did detect increased binding of CHD4 to EPS8L2 and KCNK1 promoter, and therefore both were included in several follow-up CHIP experiments (Fig. 4E- Fig. 4J). These experiments did support EPS8L2 as a ZEB1/NuRD target, however, we selected to study TBC1D2b due to the association of TBC1D2b/Rab22 axis with lung cancer progression.

Additional experiments were performed to validate the relevance of the three hits (TBC1D2a, TBC1D2b, and EPS8L2) as regulators of lung cancer invasion and have been added to the revised manuscript (Suppl. Fig. 6). Briefly, TBC1D2a, TBC1D2b and EPS8L2 were expressed in the murine cell line 344SQ (Suppl. Fig. 6A and 6B confirms overexpression by comparison of mRNA and protein to empty vector control). Transwell assays were performed to determine the effect of each gene to the invasive phenotype of 344SQ (Suppl. Fig. 6C). We found that all three genes consistently downregulated *in vitro* migration and invasion. Future experiments will be required to determine the significance of EPS8L2 in NSCLC metastasis and mechanism of action, but these are beyond the scope of this manuscript.

10. Figure 5B, legend: what is the 5% lane?

We have revised the Fig. 5B legend to clarify that the 5% control lane is 25 μ g of the whole cell lysate or 5% of the total amount of protein utilized per IP (500 μ g).

11. Figure 6B: the molecular weight shift of E-cadherin from phosphorylated to non-phosphorylated is not obvious. Controls before and after phosphatase treatment, or else, should be shown to indicate the size difference.

We have performed phosphatase treatment of 344SQ-TBC1D2b lysate to determine whether the faster migrating band observed upon TBC1D2b overexpression is in fact de-phosphorylated E-cadherin and included these results in the revised Supplemental Fig. 8D. We found that the addition of lambda phosphatase- which has activity towards phosphorylated serine, threonine and tyrosine residues- did produce a molecular weight shift equivalent to the observed molecular weight (~97 kDa), confirming that TBC1D2b does promote the accumulation of dephosphorylated E-cadherin. We have also updated the figure annotations in the manuscript to distinguish the molecular weight differences of E-cadherin.

12. Figure 6D: HOECHST not HOESCHT.

This error has been corrected in the revised manuscript.

13. Legend to Figure 6: D is missing and thus shifts E and misses F.

This error has been corrected in the revised manuscript.

Reviewer #3:

1. There should be a more complete description of where these targets comprising 235 epigenetic regulators were originally designed. Further, a description of how the 10 shRNAs/target were defined would be meaningful for a reader attempting a similar experiment.

We thank the reviewer for the suggestion, which has improved the clarity of the revised manuscript. The custom library constituting 235 genes was focused on chromatin remodeling enzymes and was originally described in Carugo et. al. 2016. Additional description of this work has been incorporated in the revised manuscript. Specifically, the following sentence has been added to the section "*Loss-of-function screen identifies epigenetic vulnerabilities in NSCLC mouse model*" to illustrate the types of targets comprising the 235 epigenetic regulators: "Target regulators included subunits of various complexes that remodel nucleosomes, catalyze post-translational modifications, deposit histone variants and methylate DNA." In addition, the following has been added to the methods section to describe the design of the shRNAs: "A custom shRNA library targeting 235 epigenes (10 independent shRNAs/gene) was constructed by using chip-based oligonucleotide synthesis and cloned into the pRSI16 lentiviral vector (Cellecta) as a pool. Targeting sequences were designed using a proprietary algorithm (Cellecta). The oligo corresponding to each shRNA was synthesized with a unique molecular barcode (18 nucleotides) for measuring representation by NGS".

2. For Fig 1B and 1C, the pool A and B are defined as biological replicates in the text of the manuscript. That should also be defined in the figure legend. Further, the error bars should be defined in the figure legend for 1B. How many technical replicates and what do the bars indicate?

Three technical replicates were performed and error bars indicate the average of three experimental runs. This detail has been incorporated throughout the manuscript.

Additionally, the subsequent sentence has been added to the figure legend of Fig. 1B in the revised manuscript to indicate the significance of the pools: "Subsequent to selection cell lines were divided into two pools (denoted Pool A or B) to reflect biological replicates."

3. The methods section for "Quantitative Real-Time PCR analysis" should be expanded to describe the methods for generating a cDNA library following the RNA isolation. Also, this section should be linked to a table in the supplemental section describing the sequence of the primers that were used.

The following sentence has been added to the methods section "Quantitative Real-Time PCR analysis" in the revised manuscript: "Total RNA was isolated using TRIzol® Reagent (Life Technologies) according to the manufacturer's instructions and reverse transcribed using qScript™ Reagent (Quanta Biosciences)."

Additionally, Supplemental Table 4 has been provided to describe the sequences of the qPCR and ChIP primers used throughout the manuscript.

4. The body of the text describes the use of a dual-luciferase reporter system. The authors should describe in the figure legend or the figure itself if the 'relative luciferase' is actually normalized firefly/renilla luciferase that is then set as a fraction to the control treatment to describe the change in activity.

Indeed, Fig. 3F expresses the firefly/renilla luciferase activity relative to the control treatment. To reflect this, "Relative luciferase activity" was modified to "Relative luciferase activity (Fold Change)" and the legend has been revised to include the following experimental details: "Relative normalized luciferase activities from reporter constructs of either empty vector (pGL3), or 321 base pair upstream fragment of miR-200b-a-429 promoter [49]. Vectors were transfected in H358-GFP control or ZEB1 cells which were pre-transfected with siRNA control or siRNA targeting CHD4. Doxycycline induction was then conducted prior to quantification of luciferase activity. Graphs represent normalization to control"

REVIEWERS' COMMENTS:

Reviewer #1 (Remarks to the Author):

The authors have satisfactorily addressed my concerns

Reviewer #3 (Remarks to the Author):

Thank you for the responses to the original review. No further comments.

Reviewer #4 (Remarks to the Author):

Manshouri and colleagues report in the present manuscript that the EMT-inducing transcription factor ZEB1 recruits the NuRD complex in NSCLCs and identify the GTPase activating protein TBC1D2b as a ZEB1/NuRD complex target gene. They further show that downregulation of TBC1D2b contributes to the endocytosis and degradation of E-cadherin, thereby inducing EMT.

The three reviewers of the original manuscript emphasized that it was scientifically sound, thoughtfully designed and mostly convincing in its conclusions. Nevertheless, they addressed several experimental issues and concerns, leading to significant changes that are underlined in the revised version of the manuscript. In particular, all the 13 points raised by Reviewer#2 are properly addressed by the authors, including the effects of HDAC inhibitors on the TBC1D2b/Rab22 axis and the rationale for the selection of TBC1D2b for further functional studies (a point also raised by Reviewer#1). In conclusion, based on the overall quality of the manuscript, the novelty of the findings and considering the responses to Reviewer's comments, I recommend publication of the manuscript in its present form.